# Evaluation of regional climate features over Antarctica in the PMIP past1000 experiment and implications for 21st-century sea level rise

Vincent Charnay [1], Daniel P. Lowry [1,2], Elizabeth D. Keller [1,2], and Abha Sood [3]

[1]Antarctic Research Centre, Victoria University of Wellington, Wellington, New Zealand
[2]Department of Surface Geosciences, GNS Science, Lower Hutt, New Zealand
[3]Centre of Sustainability, University of Otago, Dunedin, New Zealand

**Correspondence:** Vincent Charnay  (vincent.charnay@vuw.ac.nz)

**Abstract.** Surface mass balance (SMB) of the Antarctic Ice Sheet (AIS) is an important contributor to global sea level change. We look to the Last Millennium (850-1850 CE) as a period of relative climate stability to understand what processes control natural variability in SMB, distinct from anthropogenic warming. With evidence for large regional differences in climate and SMB trends over the Last Millennium in Antarctic ice core proxy records, model simulations need to be validated over such timescales to assess if they capture those regional variations in order to have confidence in end-of-century SMB projections. In this study, we provide a quantitative evaluation of paleo-simulations in simulating Last Millennium regional climate changes in Antarctica. We evaluate model performance by comparing available Paleoclimate Modelling Intercomparison Project (PMIP) past1000 models and the CESM Last Millennium Ensemble (CESM-LME) to four sets of Last Millennium Antarctic proxy-based reconstructions that are most relevant to the SMB: snow accumulation, surface air temperature (SAT), sea surface temperature (SST) and Niño 3.4 index, using a multi-parameter scoring method. Our results show that no single model performs consistently well across all variables. Models have reasonable strength in capturing SATs and SSTs, while showing strong biases for both snow accumulation and the Niño 3.4 index. The best-performing model, CESM-LME, predicts higher SMB by 2100, which implies stronger mitigation of the projected dynamic ice loss contribution of the AIS to sea level rise.

## 1 Introduction

The surface mass balance (SMB) of the Antarctic Ice Sheet (AIS), defined as the balance at the surface of the ice sheet between accumulation, in the form of precipitation, and ablation, in the form of surface runoff, sublimation and blowing snow erosion (Lenaerts et al., 2019), is important for its influence on sea level (Ligtenberg et al., 2013). An increase in snowfall accumulation over the AIS is believed to have mitigated twentieth-century sea-level rise (Medley and Thomas, 2019). However, the large range of natural climate variability makes it difficult to determine if this is due to short-term fluctuations in precipitation or a longer-term trend driven by anthropogenic change (Lenaerts et al., 2019). Projections of 21st-century SMB span a large range and involve uncertainties derived from insufficient understanding of processes important for polar climate and structural differences among climate models (Li et al., 2023). In most parts of the Antarctic continent, SMB is expected to increase as a result of enhanced snowfall in response to atmospheric warming (Frieler et al., 2015; Lenaerts et al., 2019), while the runoff and surface melt remain small (Winkelmann et al., 2012; Ligtenberg et al., 2013; Lenaerts et al., 2016). Influences on

SMB include large-scale atmospheric circulation and ocean conditions, as well as small-scale topographic features, making it challenging to model.

Numerous studies have evaluated climate models in their ability to simulate Antarctic climate features related to SMB over the historical period to help refine future projections (Agosta et al., 2015; Palerme et al., 2017; Gorte et al., 2020). For example, Gorte et al. (2020) found that models which best captured reconstructed historical SMB, based on mean value, trend, temporal variability, and spatial distribution, tended to project smaller SMB increase by the end of the century. In contrast, Palerme et al. (2017) showed that models which compare best with observed historical snowfall tended to project larger snowfall increase into the 21st century. These different results highlight the importance of how model performance is evaluated, and the potential limit of focusing on the historical period for understanding those future end-of-century changes. Hence, if we want to improve confidence and predict credible end-of-century SMB, we need a longer time period to compare against. Other studies have noted the importance of going beyond the historical period and looking at past climate using proxy-based reconstructions to assess model skill (Hargreaves et al., 2013; Schmidt et al., 2014; Bracegirdle et al., 2019). Performing a model-proxy comparison provides us with an opportunity to evaluate the performance of climate models in simulating climate features over a time period that is commensurate with projected future changes (Hargreaves et al., 2013).

The Last Millennium (LM, 850-1850 CE) is a climate state of relative stability (Bradley et al., 2003; Jones et al., 2001), making it an important period for past climate research by providing the opportunity to study the variability and response of Earth's climate to small shifts in climate forcings and by separating anthropogenic impacts from natural climate variability (Jungclaus et al., 2017). The LM is therefore a useful candidate to understand natural variability without having to disentangle the signal from anthropogenic warming. The LM is primarily divided into two periods, the Medieval Climate Anomaly (MCA, 850-1350) associated with warmer global temperatures, and the Little Ice Age (LIA, 1350-1850) a period of relatively colder global temperatures (Hughes and Diaz, 1994; Bertler et al., 2011; Rhodes et al., 2012).

Recent LM temperature reconstructions find no evidence of a globally coherent warmer MCA over Antarctica (Neukom et al., 2019; Perkins and Hakim, 2021), but rather a long cooling across both MCA and LIA (Stenni et al., 2017). There is growing evidence from Antarctic ice core records for large regional differences in SMB trends over the LM (Thomas et al., 2017), with notably long-term negative trends —centennial-scale— over the West Antarctic Ice Sheet and Victoria Land coast, and long-term positive trends over the Antarctic Peninsula and Weddell Sea and Dronning Maud Land coastal regions. The actual drivers for such regional variations remain uncertain, demonstrating the need for a regionally focused study. Assessing general circulation model (GCM) skills in simulating LM climate features can guide model development in capturing drivers of regional SMB variability on a finer scale.

The LM is among the periods selected by the Paleoclimate Modelling Intercomparison Project Phase 3 and 4 (PMIP3 and PMIP4) for experiments contributing to the Coupled Modelling Intercomparison Project Phase 5 and 6 (CMIP5 and CMIP6) (Jungclaus et al., 2017). The goals of the pre-industrial millennium PMIP experiments (past1000, 850-1850 A.D.) are to study the response to natural forcing under stable climate and conditions resembling those of the current climate. The relatively abundant proxy data available in some regions makes the LM a valuable period for evaluating model skills in capturing regional climate features, as this broader spatial coverage allows for a better understanding of regional trends (Cook et al., 2008;

PAGES2k, 2013; Thomas et al., 2017; Stenni et al., 2017). The uneven regional distribution of the data will allow us to constrain some regions better than others. PMIP past1000 models can be validated over long timescales to assess if they capture those regional variations.

To this end, we examine the model skill of the PMIP model ensemble with a specific focus on variables that influence and are important to simulate SMB accurately. We build on the scoring method in Gorte et al. (2020) to evaluate the PMIP3/4 models that participated in the past1000 experiments for which data is publicly available, using quantitative Antarctic paleoclimate reconstructions as observations. For this, two objectives were identified, including (1) evaluation of model ability to simulate regional climate changes; (2) multi-parameter evaluation of overall model skill. We discuss model biases, strengths and weaknesses and compare results obtained with historical simulations in Gorte et al. (2020). We also use this scoring method to guide the selection of models for Regional climate model (RCM) forcing.

## 2  Data

### 2.1  PMIP models

We assess all PMIP past1000 models for which SAT, SST and snow accumulation (precipitation - evaporation) are available, as well as the Community Earth System Model (CESM) Last Millennium Ensemble, for a total of twelve models, including eight PMIP3 models (MRI-CGCM3, MIROC-ESM, MPI-ESM-P, CSIRO-Mk3L-1-2, GISS-E2-R, BCC-CSM1-1, HadCM3 and the Community Climate System Model version4 (CCSM4)) (Watanabe et al., 2011; Gent et al., 2011; Yukimoto et al., 2012; Phipps et al., 2012; Giorgetta et al., 2013; Wu et al., 2013; Schmidt et al., 2014; Valdes et al., 2017; Gutjahr et al., 2019), three PMIP4 models (MRI-ESM2-0, MIROC-ES2L and ACCESS-ESM1-5) (Yukimoto et al., 2019; Hajima et al., 2020; Ziehn et al., 2020), and the CESM-LME model (Otto-Bliesner et al., 2016). Additional PMIP past1000 models are excluded from this analysis due to our initial selection criteria (see Section 2.2). The resolutions and numbers of vertical layers for both the atmosphere and ocean are shown in Table 1.

The past1000 simulations serve to investigate the response to mainly natural forcing under background conditions resembling those of the current climate, i.e. the pre-industrial millennium. These simulations are based on a common protocol (Schmidt et al., 2011; Jungclaus et al., 2017), describing a variety of suitable forcing boundary conditions, such as orbital parameters, solar irradiance, stratospheric aerosols of volcanic origins, and atmospheric greenhouse gas concentrations. The changes between the common protocol for PMIP3 and PMIP4 past1000 simulations are mostly derived from the use of newly available records, permitting a more comprehensive reconstruction of external forcing.

The CESM-LME employs version 1.1 of CESM with the Community Atmosphere Model version 5 (CESM1-CAM5) (Otto-Bliesner et al., 2016). The CESM-LME provides the largest ensemble of LM simulations with a single model to date, including a total of 13 members for the full forcing experiment. The only difference between ensemble members is a small (order 10e-14) random roundoff difference in the air temperature field at the start of each simulation. The forcing over the LM includes orbital, solar, volcanic, changes in land use/land cover and greenhouse gas levels, and their implementation follows those used in PMIP3 (Otto-Bliesner et al., 2016).

**Table 1.** Atmospheric and oceanic model resolutions of the PMIP models analysed in this study, along with their respective numbers of vertical layers.

| Models | Atmosphere | | Ocean | | |
| --- | --- | --- | --- | --- | --- |
| | Horizontal (°) | Vertical (nb layers) | Horizontal (°) | Vertical (nb layers) | PMIP phase |
| MRI-ESM2-0 | 1.125 x 1.125 | 80 | 0.5 x 1 | 60 | PMIP4 |
| MIROC-ES2L | 2.8125 x 2.8125 | 40 | 1.4 x 1.4 | 62 | PMIP4 |
| ACCESS-ESM1-5 | 1.25 x 1.875 | 38 | 1 x 1 | 50 | PMIP4 |
| MRI-CGCM3 | 1.125 x 1.125 | 48 | 0.5 x 1 | 50 | PMIP3 |
| MPI-ESM-P | 1.875 x 1.875 | 47 | 1.5 x 1.5 | 40 | PMIP3 |
| MIROC-ESM | 2.8125 x 2.8125 | 80 | 1.4 x 1.4 | 44 | PMIP3 |
| CSIRO-Mk3L-1-2 | 3.18 x 5.625 | 18 | 3.18 x 5.625 | 21 | PMIP3 |
| GISS-E2-R | 2 x 2.5 | 40 | 1 x 1.25 | 32 | PMIP3 |
| BCC-CSM1.1 | $2.8125 \times 2.8125$ | 26 | 1 x 1 | 40 | PMIP3 |
| HadCM3 | 3.75 x 2.5 | 19 | 1.25 x 1.25 | 20 | PMIP3 |
| CCSM4 | 1.25 x 0.9 | 26 | 1 x 1 | 60 | PMIP3 |
| CESM-LME | 2.5 x 2.5 | 70 | 1 x 1 | 60 | - |

## 2.2 Paleoclimate proxy records

Our knowledge of past Antarctic climate trends comes predominantly from a combination of proxy records from natural archives and paleoclimate models. To assess climate model performance, we rely on proxy records of Antarctica's climate and Southern Ocean conditions. We assess model skill by comparing the model outputs with four proxy-based reconstructions that are most relevant to the SMB: snow accumulation, surface air temperature (SAT), sea surface temperature (SST) and Niño 3.4 index. Other variables are also important for these processes but we are constrained by what reconstructions are available.

The past snow accumulation dataset is compiled by the PAGES Antarctica2K working group (Thomas et al., 2017), which presents annual Antarctic snow accumulation variability at the regional scale over the past 1000 years. The dataset is comprised of 79 Antarctic ice core records, 44 of which cover the LM period (Figure 1). The estimates of snow accumulation are based on the physical distance between suitable age markers (bulk changes in isotopic composition reflecting glacial cycles, volcanic eruptions for decadal to millennial timescales, seasonal variations in stable water isotopes, and chemical species including sea salts, hydrogen peroxide, radio isotopes, and biologically controlled compounds within the ice core (Dansgaard and Johnsen, 1969). These snow accumulation reconstructions provide valuable information on changes in certain regions; however, poor spatial coverage in some regions may result in misleading regionally-averaged trends. Thomas et al. (2017) suggest that a greater spatial representation with a higher number of ice core records, especially in the East Antarctic Plateau and Weddell

Sea coastal regions, will improve the understanding of the true nature of Antarctic SMB in the past. We thus compare the reconstructed snow accumulation to GCMs output for each ice core record individually.

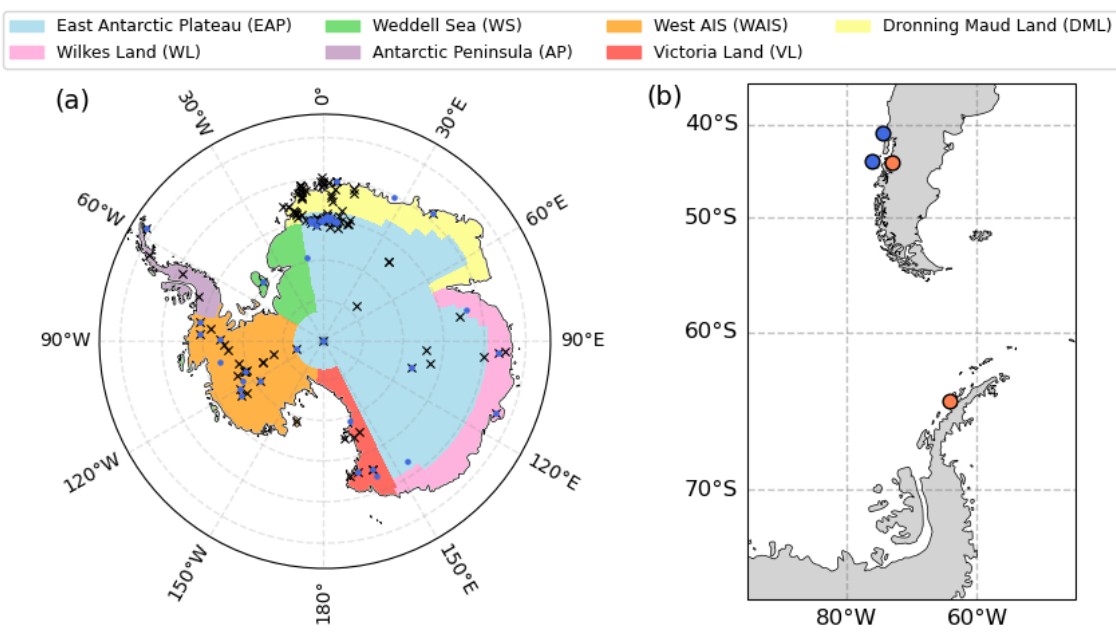

**Figure 1.** (a) Locations of ice core sites with reconstructed SAT (black crosses) and snow accumulation (blue dots) (Thomas et al., 2017; Stenni et al., 2017) and the SAT regional boundaries from Stenni et al. (2017) used in this study. (b) Sediment core locations for the Southern Ocean sea surface temperature reconstructions for annual (blue) and seasonally averaged over the austral spring (orange) (PAGES2k, 2013).

The surface air temperature (SAT) is obtained from a database compiled by the PAGES Antarctica2k working group (Stenni et al., 2017). Paleotemperatures are reconstructed based on the statistical relationship between $\delta^{18}O$ of water/precipitation and SAT. The database consists of 112 ice core records, shown in Figure 1, which are temporally resolved at a 10-year average and reconstruct the last 2000 years. The reconstruction only provides regionally averaged LM SAT anomalies (referenced to the 1900–1990CE period) time series over seven Antarctic regions: the East Antarctic Plateau (EAP), Wilkes Land coast (WL), Weddell Sea coast (WS), Antarctic Peninsula (AP), West AIS (WAIS), Victoria Land coast (VL), and Dronning Maud Land coast (DML) (Figure 1). In addition to these seven regions, there are also reconstructions for a continent-wide Antarctic region, broad-scale West and East Antarctic. Uncertainties arise from the uneven representation of ice core spatial coverage (EAP and WS) and the relative weak covariance on average between $\delta^{18}O$ and SAT (Klein et al., 2019).

In addition to Antarctic climate records, we use reconstructions of Southern Ocean surface conditions. The PAGES Ocean2k (PAGES2k, 2013) group provides 57 SST reconstructions across the global ocean. Cores of sediment accumulated on the seafloor create excellent past archives and are used to reconstruct past ocean changes (Moffa-Sánchez et al., 2019). Here, we focus on the four reconstructions located in the Southern Ocean (Figures 1). Of these, two reconstructions are annual,

and the other two are seasonally averaged over the austral spring (SON). There are two types of proxies with their respective calibration; Alkenones with the PRA1988 calibration (Prahl et al., 1988) and TEX86 with the KIM2008 calibration (Kim et al., 2008). The main uncertainty of those reconstructions is the relative low temporal resolution with decades-scale gaps.

Teleconnections arising from the El Niño-Southern Oscillation (ENSO) play a crucial role in shaping recent Antarctic climate trends and SMB (Lüning et al., 2019). ENSO exercises an influence on Antarctic climate by weakening or strengthening the Amundsen Sea Low, depending on its phase, which directly influences the atmospheric moisture over West Antarctica and, subsequently, the amount of precipitation (Ding et al., 2011; Clem et al., 2018). As a result, we include the ENSO index in the scoring. The ENSO index reconstruction is based on tree-ring data from Mexico and Texas, USA (Cook et al., 2008). The tree-ring is a natural archive of past climate and has been widely used notably for its high temporal resolution and accuracy of dating (Hughes, 2002). The dataset provides a reconstruction of Niño 1+2 (0-10S, 90W-80W), 3 (5N-5S, 150W-90W), 3.4 (5N-5S, 170W-120W), and 4 (5N-5S, 160E-150W) indices over the past 700 years, extending back to 1300 CE, with the best verified portion beginning in 1400 CE. In this study, we focus on the Niño 3.4 reconstruction as it is the most commonly used index to define El Niño and La Niña events.

## 3  Methods

To evaluate model outputs against the SAT, snow accumulation (defined here as precipitation minus surface evaporation-sublimation (P-E)) and SST LM time series, we use the method developed by Gorte et al. (2020), which outlines three criteria on which to score the time series variables — mean, trends, and variability. Despite uncertainties, the reconstructions offer robust and valuable information, and the method we use for the evaluation explicitly accounts for these uncertainties, supporting their use in assessing model performance (Klein et al., 2019; Gorte et al., 2020). The mean value is evaluated by giving a score $x$, based on how many $x$ times the reconstructed uncertainty (defined here as $\pm 1\sigma$) is required for the entire time series to be within the reconstructed uncertainty. Models with a closer time series mean to the reconstructed mean will then be attributed a better score, with a score of 1 being the best. Similarly, the time series trend score $y$ is the multiple of the reconstructed trend uncertainty required to capture the model trend. Lastly, the temporal variability is calculated on normalised time series to avoid double-counting the impact of SMB mean value (because this is already covered by the first scoring criteria). A score $z$ is given for how many $z$ times the normalised reconstruction standard deviation was required to capture the normalised model standard deviation. For the SAT, the three criteria are assessed on regionally averaged time series, and for the snow accumulation and SST, we apply the criteria on a pixel-to-ice core comparison, by extracting the model pixel corresponding to the location of the ice core (Table S1 and S2). For CESM-LME, the use of the ensemble mean time series will form temporal variability biases. Therefore, the CESM-LME score will be the average of all 13 ensemble members' scores.

For the Niño 3.4 index, the focus is not on whether a particular model reproduces a particular El Niño or La Niña event, but rather to determine if the model simulates a similar number of events over a given time period of X years, here 1400-1850 CE. We identify an El Niño and La Niña event with a threshold of +/- 0.4°C. Hence, we calculate the absolute difference of the number of El Niño and La Niña occurrences in the model output with occurrences from the reconstruction over the period of

X years. The score is the addition of both El Niño and La Niña differences, with the smallest score indicating that the model that simulates the Niño 3.4 index the best.

We normalised each set of scores to be on a scale from 1 to 10 to ensure that each criterion was equally weighted. The total score is the average of all sets of normalised scores. The score is an indication of the model's performance in comparison to all the other models. Smaller total scores indicate stronger model performance and higher scores indicate poorer model performance.

## 4 Results

### 4.1 Snow accumulation

Overall the models show poor skill in simulating the snow accumulation over the AIS. Figure 2 shows the normalised regional snow accumulation score, calculated by averaging the score of all ice cores within each region. This highlights the spatial variability in snow accumulation and helps identify potential regional biases. The most consistent model across all regions is CSIRO-Mk3L-1-2 and is the best scoring model in the AP and WS while maintaining a score of 3 or below everywhere else. CESM-LME mean is the best scoring model in the EAP, VL and WAIS but scores in the bottom half of models in the WL. CCSM4 shows the same strengths and weaknesses as CESM-LME mean but performs slightly worse for all of them. MRI-ESM2-0 shows strength in the East Antarctic coastal region, as it is the best scoring model in both the WL and DML and maintains a score of 5 or below everywhere else. ACCESS-ESM1-5 performs well in the WAIS and the EAP, but scores in the bottom half of models in the WL. MPI-ESM-P performs relatively well for most regions but shows snow accumulation regional biases in the AP. MIROC-ES2L shows regional biases in the EAP, WAIS and VL as it consistently overestimates accumulation.

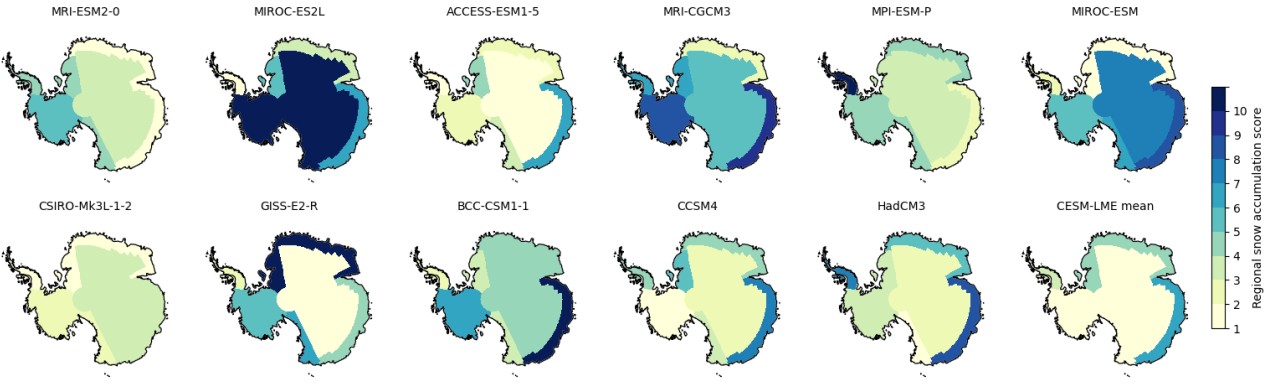

**Figure 2.** Normalised regional snow accumulation score of seven regions (EAP, WL, WS, AP, WAIS, VL and DML) over the LM. All ice core scores within one region are averaged together following Thomas et al. (2017) regional boundaries, which are slightly different than those in Stenni et al. (2017) (used in Figure 1). The best score is 1 (off white), and the worst score is 10 (dark blue). The score is an indication of the model's performance in comparison with other models.

Figure 3 shows the mean, trends and temporal variability values of reconstructed and modelled time series for each ice core location allocated over their respective seven Antarctic regions. Details of the ice core records are shown in Table S1. Overall, the models show poor skill in simulating the snow accumulation over the AIS. They tend to overestimate the snow accumulation mean, while not capturing the trends and magnitudes of temporal variability. The greatest accumulation rates occur in the AP, WL and WAIS, while in the interior, VL and WS show modest accumulation rates.

Ten ice core records are located in the WAIS region. CESM-LME, MPI-ESM-P and ACCESS-ESM1-5 are models with mean accumulation within the reconstructed uncertainty for most proxies, while MIROC-ES2L, MIROC-ESM and MRI-CGCM3 consistently exhibit greater accumulation. No models succeed in capturing the signs and magnitudes of trends for locations with trends larger than $1.0\ \mathrm{mm\ yr^{-2}}$. For locations with modest trends, models generally agree. CESM-LME performs best in terms of capturing temporal variability, while other models underestimate it.

Three ice core records are located in the AP region. CSIRO-Mk3l-1-2, MIROC-ES2l and BCC-CSM1-1 are always within the mean reconstructed uncertainty, and CSIRO-Mk3l-1-2 is the only model that captures the correct signs and magnitudes of trends for all three sites. MPI-ESM-P underperforms in this region and displays large differences in the trends and temporal variability.

Two ice core records are located in the WL region. Models underestimate the accumulation and fail to capture the significant trend of the second proxy, a recurrent issue in this study. The model that manages the best is MRI-ESM2-0, for both the mean and trends.

Only one proxy record exists for the WS. MIROC-ESM, MPI-ESM-P and BCC-CSM1-1 are the only models within the mean reconstructed uncertainty. Models agree with the modest trend but fail to be within the trend uncertainty and only CSIRO-Mk3L-1-2 captures the temporal variability. GISS-E2-R and MRI-CGCM3 exhibit the largest discrepancies in accumulation compared to the reconstruction.

Four ice core records are located in the VL region. CESM-LME scores the best for all three criteria. MIROC-ES2L shows potential regional bias as it displays the largest differences for mean, trends and temporal variability.

The EAP has the largest number of records with 21 ice core records, but most are located near the coast in close proximity to DML, making EAP poorly represented spatially. Proxies exhibit large annual temporal variations, suggesting that the annual accumulation rates vary substantially. This is not surprising considering that accumulation in this region is so low that even a small absolute increase in accumulation means a large relative increase (Frieler et al., 2015). CESM-LME, GISS-E2-R, ACCESS-ESM1-5 and HadCM3 are the four models that best capture the mean. Models in general struggle to capture the trends, but ACCESS-ESM1-5 and MPI-ESM-P perform the best. Similarly, no models consistently reproduce the large temporal variabilities, but CESM-LME is consistently the closest. MIROC-ES2L exhibits a strong regional bias in the EAP as it shows the largest differences for all three criteria.

Because the time series only covers the last 100 years of the LM, the reconstructed uncertainties from the three ice core records in the DML, especially for the trends uncertainties, are much larger. This means that, despite apparent differences in sign and magnitude, most models fall within the uncertainty range of the reconstruction.

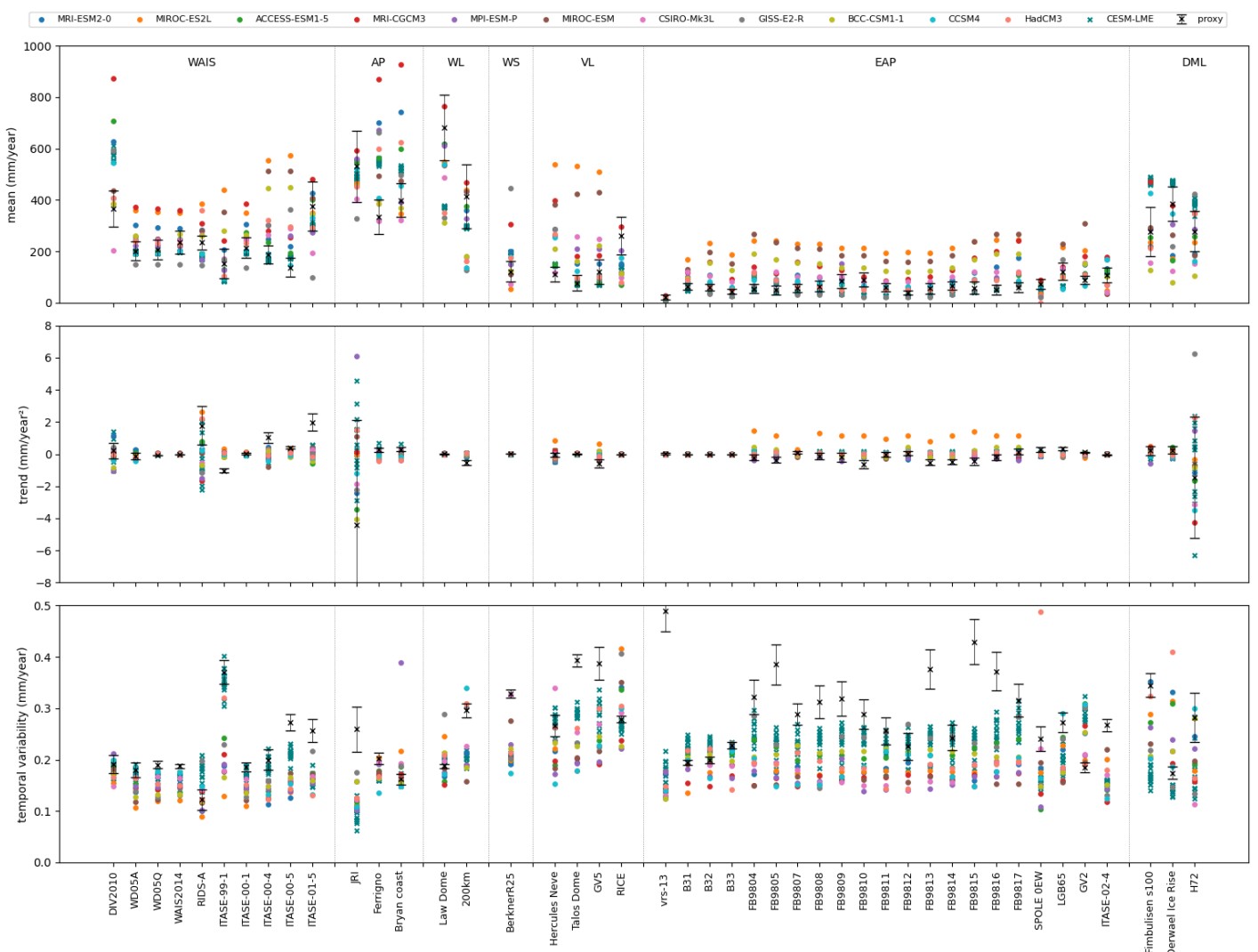

**Figure 3.** Comparison of simulated and reconstructed mean, trends and temporal variability snow accumulation over the LM for each ice core record (Table S1). Each ice core record is regrouped in seven Antarctic regions — WAIS, AP, WL, WS, VL, EAP and DML.

## 4.2 Surface air temperature

In contrast with snow accumulation, the models show agreement with the reconstruction in simulating SAT over the AIS. MRI-CGCM3 is the best-scoring model over five regions but is ranked last in the DML (Figure 4). While ACCESS-ESM1-5, GISS-E2-R and MIROC-ES2L are the top three scoring models for the DML, they score relatively poorly for the rest of the continent. It is important to note, however, that because the ice core records do not cover the full LM in the DML, this lack of temporal representation makes it difficult to rigorously assess the performance of the models in this region. Other noteworthy

models are the CESM-LME mean, which is among the best-scoring models over six regions, and MPI-ESM-P, which scores
in the bottom half of models only in the AP and DML. The warm bias of MIROC-ESM is reflected in its regional normalised
score, as it is the worst-performing model in six regions out of seven and the second-worst in the seventh region. Despite its
overall warm bias, MRI-ESM2-0 shows a better score than the overall model average over five regions (AP, WS, EAP, DML
and WL), due to scoring better for the temporal variability criteria compared to the other models.

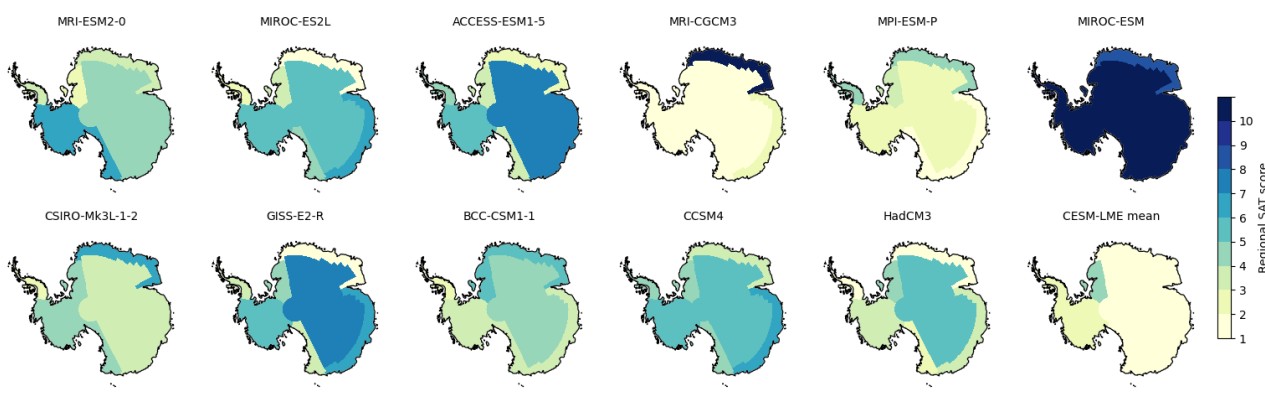

**Figure 4.** Normalised regional SAT score of seven regions (EAP, WL, WS, AP, WAIS, VL and DML) over the LM. The best score is 1 (off
white), and the worst score is 10 (dark blue). The score is an indication of the model's performance in comparison with other models.

Figure 5 shows time series of the regionally averaged SAT anomalies for both the reconstruction and model simulations. Ice
core reconstructions suggest a slight broad-scale cooling trend over most of continental Antarctica, with modest statistically
significant temperature decreases over four regions: the EAP, WAIS, VL, and WL. The WS and AP do not display any statis-
tically significant trends, while the DML shows the greatest temperature change. In contrast to these records, MIROC-ESM
and MRI-ESM2-0 show positive trends for all regions, with the former starting to show temperature increases at the 1000 CE
mark, suggesting that MIROC-ESM has a warm bias. For MRI-ESM2-0, the warm bias is more modest in magnitude. All other
models are consistent with the general broad-scale cooling trend and generally show similar trend magnitudes.

In terms of the models consistent with the sign of change indicated by the reconstructions, only ACCESS-ESM1-5 and
MRI-CGCM3 show a positive trend in the DML. Models show a slight cold bias in the WAIS and WL as the SAT anomalies
are slightly lower in the models compared to the reconstructions. In each region, the main discrepancy between the models and
the regionally averaged temperature reconstructions is the temporal variability, with modelled SAT exhibiting lower variability.
Only in the WAIS region did the reconstruction also exhibit a similar low magnitude of variability.

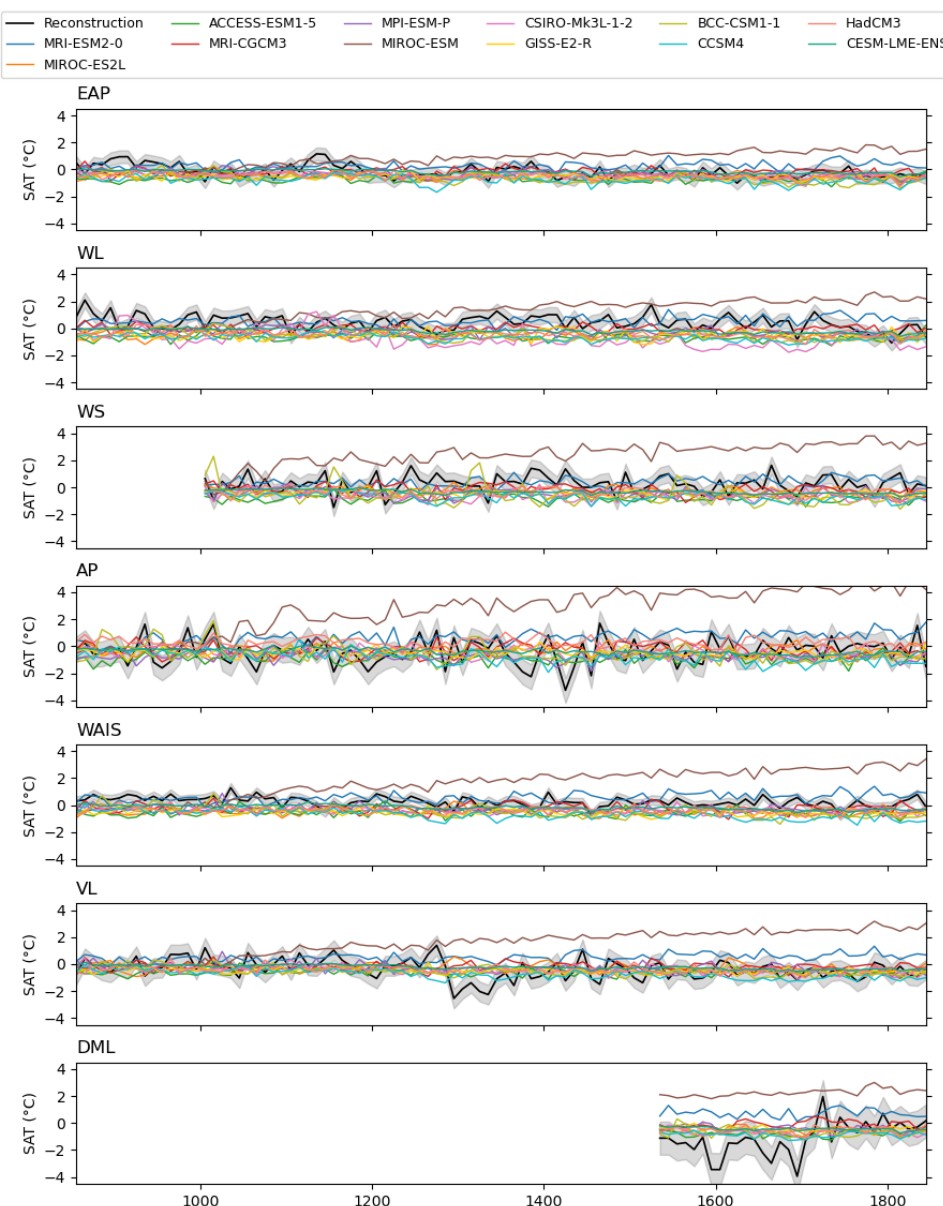

**Figure 5.** Time series of 10-year average SAT anomalies (°C) of seven regions (EAP, WL, WS, AP, WAIS, VL and DML) over the LM relative to the pre-industrial era (1900-1990 CE) of all model outputs and regionally averaged ice core temperature reconstructions. The grey shading indicates the reconstructed uncertainty (defined as $\pm 1\sigma$).

The best scoring model for the continent-wide Antarctica, West Antarctica (incorporating the AP and WAIS) and East Antarctica (incorporating the EAP, WL, WS, VL and DML) is MRI-CGCM3 (Figure 6). The second best scoring model is

CESM-LME mean. MPI-ESM-P performs slightly worse than when factoring in all the seven regions (Figure 4) but still remains better than the average. MIROC-ESM is the worst scoring model in terms of continent-wide Antarctica, West Antarctica

and East Antarctica, followed by MRI-ESM2-0 with a normalised score of 6, a 4-point score difference.

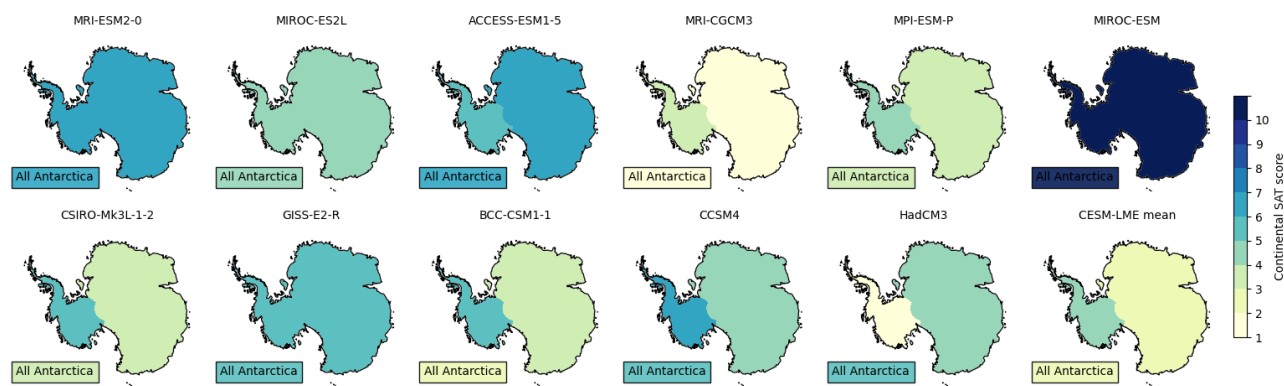

**Figure 6.** Normalised regional SAT score of three regions (Continent-wide Antarctica, West Antarctica and East Antarctica) over the LM. The best score is 1 (off white), and the worst score is 10 (dark blue). The score is an indication of the model's performance in comparison with other models. The continent-wide Antarctic score is displayed in the "All Antarctica" box on the bottom left of each Antarctic map.

Figure 7 is the same as Figure 5 but for time series averaged for the continent-wide Antarctica, West Antarctica, and East Antarctica, and shows a cooling trend in both continent-wide Antarctica and East Antarctica. Overall, the models show reasonable agreement with the reconstructions over these broader spatial scales. MIROC-ESM and MRI-ESM2-0 continue to show warm biases. Similar to the regional analysis of the seven individual Antarctic regions, models continue to show slightly colder

SAT anomalies compared to the reconstructions.

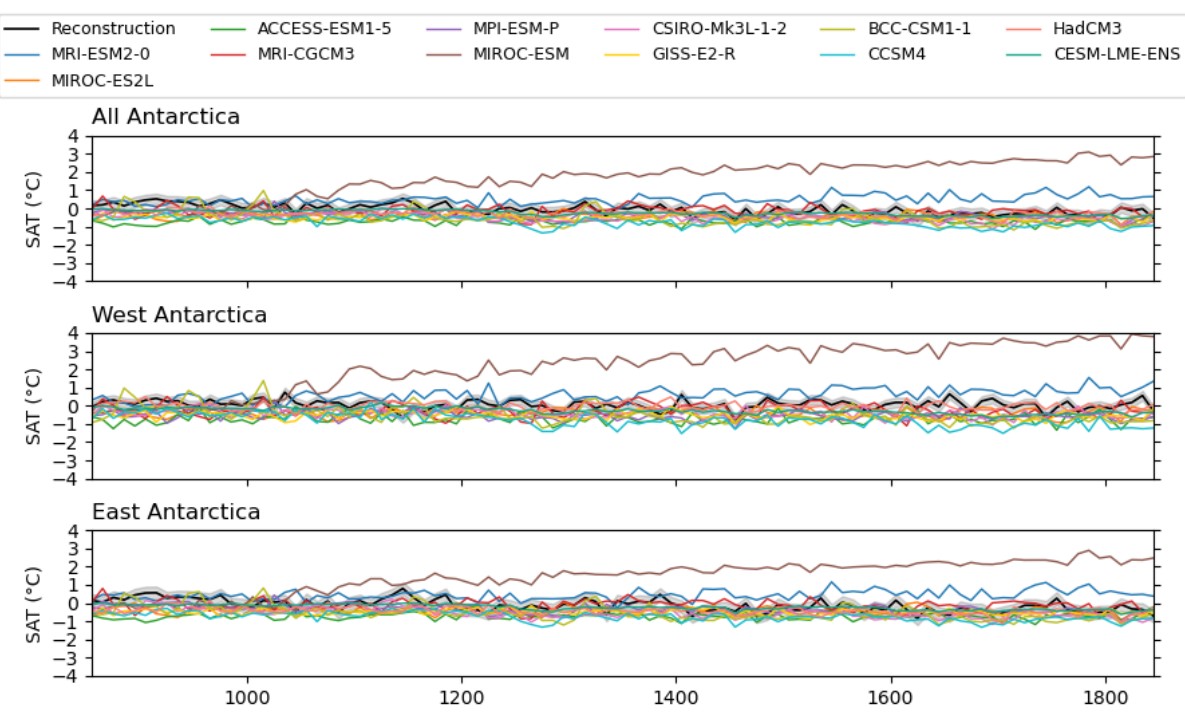

**Figure 7.** Time series of 10-year average SAT anomalies (°C) of three regions (continent-wide Antarctica, West Antarctica, and East Antarctica) over the LM relative to the pre-industrial era (1900-1990 CE) of all model outputs and ice core temperature reconstructions. The grey shading indicates the reconstructed uncertainty (defined as $\pm 1\sigma$). For the continent-wide Antarctica, the ice core reconstruction is an average of the regionally averaged EAP, WL, WS, AP, WAIS, VL and DML reconstructions. Here, West Antarctica is an average of the AP and WAIS reconstructions, and East Antarctica is an average of the EAP, WL, WS, VL and DML reconstructions.

## 4.3 Sea surface temperature

Models show skills in capturing SST trends and temporal variability but exhibit a cool SST bias in all four locations. In Figure 8, at site 1 (lat=-44.33°, lon=-72.97°), the best scoring model is MIROC-ES2L, followed by MRI-CGCM3 and GISS-E2-R. At site 3 (lat=-44.15°, lon=-75.16°), the best scoring models are ACCESS-ESM1-5, CSIRO-Mk3L-1-2 and GISS-E2-R. At site 4 (lat=-41°, lon=-74.45°), the best scoring model is MRI-CGCM3, followed by ACCESS-ESM1-5. MIROC-ESM is the worst scoring model for all sites. For the second site (lat=-64.87°, lon=-64.20°), off the western coast of the AP with SST seasonally averaged over the austral spring, all models disagree with the reconstruction and simulate SST at the freezing temperature. The models consistently simulate sea ice over that time of the year for this location, whereas the reconstruction implies the presence of sea ice only towards the end of the LM. Considering this model-proxy disagreement, we gave a score of 10 for all models.

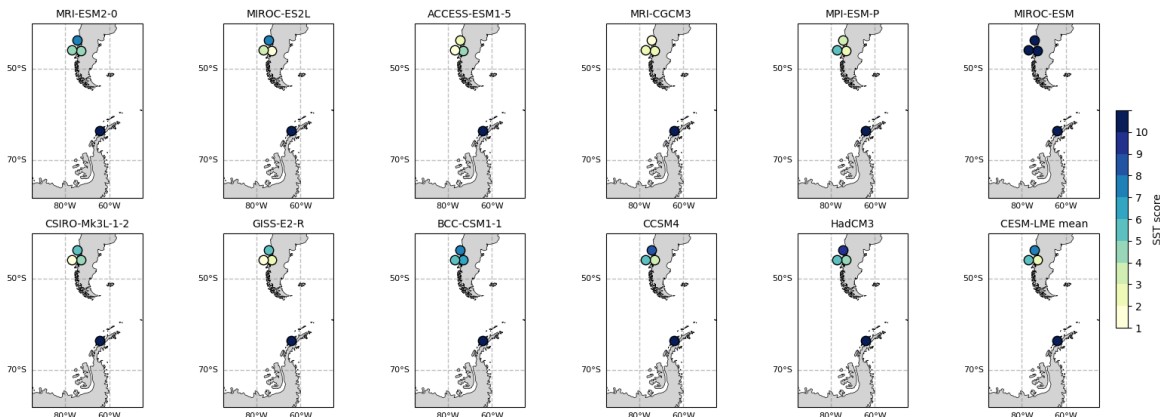

**Figure 8.** Normalised SST score of the four Southern Ocean sites over the LM. The best score is 1 (off white), and the worst score is 10 (dark blue). The score is an indication of each models performance in comparison with the other models.

SST reconstructions show a modest temperature cooling at all four marine sediment record sites (Figure 9). Only MRI-CGCM3 and ACCESS-ESM1-5 consistently show similar temperature means at site 1, 3 and 4. CESM-LME mean, MPI-ESM-P and BCC-CSM1-1 display similar signs of change and magnitude of cooling, while MRI-ESM2 and MIROC-ESM are the only models that display warming trends at all sites.

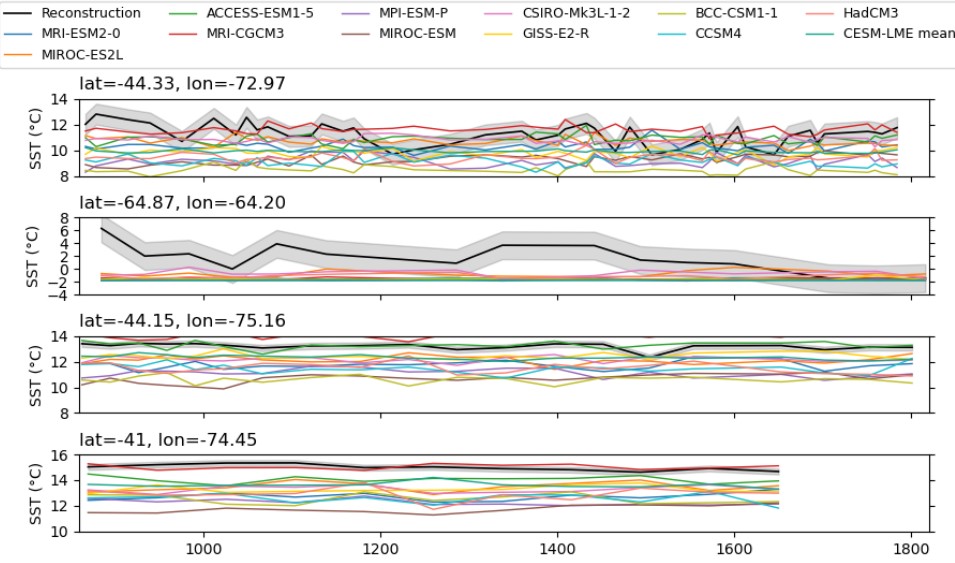

**Figure 9.** SST time series (°C) of four Southern Ocean sites over the LM of all GCM outputs and ice core temperature reconstructions. The grey shading indicates the reconstructed uncertainty (defined as $\pm 1\sigma$).

## 4.4 ENSO index

Overall, there is a strong disparity between models in their ability to represent the Niño 3.4 index. MIROC-ES2L is the closest to the reconstruction at representing ENSO (Table 2). It simulates a similar number of La Niña events and slightly fewer El Niño events than the reconstruction. CESM-LME mean and HadCM3 likewise capture well the number of La Niña events and slightly underestimates the number of El Niño events, followed closely by MRI-ESM2-0, which underestimates the number of La Niña events, but shows a similar number of El Niño events as the reconstruction. The other models are scored more
poorly with respect to ENSO. MPI-ESM-P, MRI-CGCM3, CCSM4 and CSIRO-Mk3L-1-2 produce a similar number of La Niña events to the reconstruction, but differ considerably in terms of El Niño. BCC-CSM1-1, GISS-E2-R, MIROC-ESM and ACCESS-ESM1-5 differ from the reconstruction for both number of La Niña and El Niño events.

**Table 2.** The number of El Niño and La Niña events simulated over 1400-1850 CE for each model and their respective normalised score.

| Models | number of El Niño | number of La Niña | score |
|---|---|---|---|
| Reconstruction | 92 | 100 | - |
| MRI-ESM2-0 | 85 | 76 | 1.7 |
| MIROC-ES2L | 73 | 97 | 1 |
| ACCESS-ESM1-5 | 26 | 128 | 6.4 |
| MRI-CGCM3 | 40 | 80 | 4.8 |
| MPI-ESM-P | 44 | 107 | 3.5 |
| MIROC-ESM | 65 | 26 | 7 |
| CSIRO-Mk3L-1-2 | 22 | 93 | 5.2 |
| GISS-E2-R | 16 | 123 | 6.8 |
| BCC-CSM1.1 | 11 | 160 | 10 |
| CCSM4 | 56 | 118 | 3.4 |
| HadCM3 | 70 | 92 | 1.6 |
| CESM-LME | 69 | 96 | 1.4 |

## 4.5 Total score

Figure 10 shows the total score for each model along with their respective normalised snow accumulation, SAT, SST and Niño
3.4 index scores. As described in the method section, the best score is 1. The overall skill of the paleo-simulations for the LM is uneven depending on the variable considered, as no model performs equally well for all four climate variables. The mean score across the eleven models is 4.3. Gorte et al. (2020) stated that models that score above the 90th percentile make up the subset of best scoring models. Only one model comprises this top 90th percentile — the CESM-LME mean with a score of 2.2. However, this is the mean score for CESM-LME as we averaged all 13 ensemble member scores together. The best

PMIP past1000 model is CSIRO-Mk3L-1-2 with a score of 2.9. The poorest performing models include MIROC-ESM and BCC-CSM1-1, two PMIP3 models, with respective scores of 8.2 and 6.3. The mean model score is 3.83 for PMIP4 models and 4.87 for PMIP3 models. No PMIP4 models are part of the best scoring models, but none are part of the poorest performing models (i.e. with an total score in the bottom half of models). All three PMIP4 models perform better or equal than the mean of all models, whereas PMIP3 models cover a more diverse range in scoring.

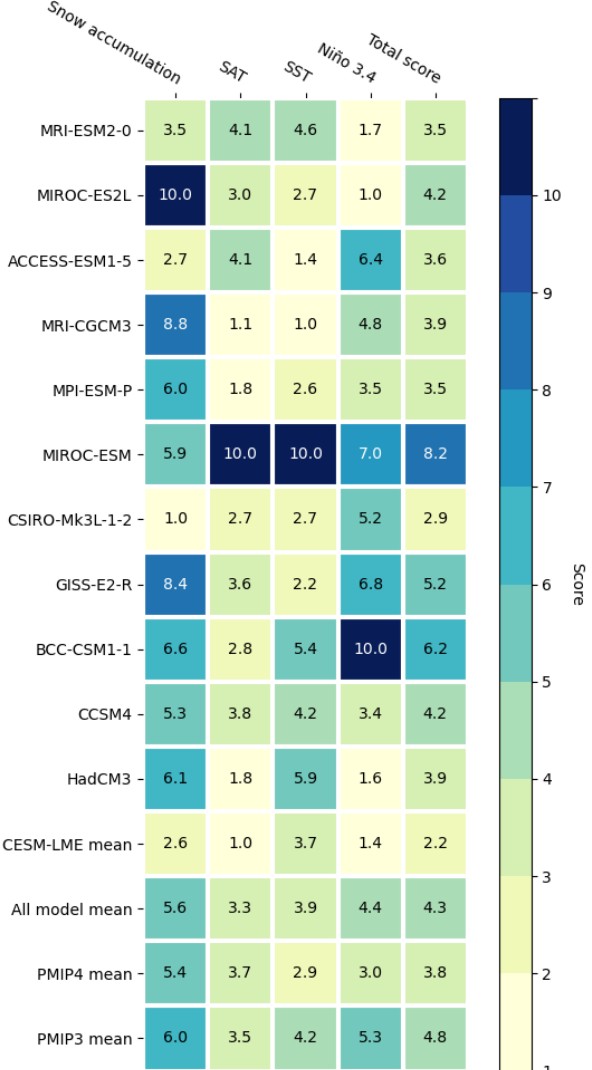

**Figure 10.** Heatmap of the normalised scores for all PMIP past1000 models and the CESM-LME mean.

## 4.6    Modelled SMB future projections

After evaluating models based on variables important for SMB over the LM, we now consider projections of AIS SMB. Similarly to Gorte et al. (2020), we defined the spatially integrated AIS SMB as precipitation minus sublimation. Here we compare the modelled AIS SMB projections between two scenarios, SSP2-4.5 and SSP5-8.5 (Figure 11). Of all the models we evaluated, future scenarios were not available for three models, CSIRO-Mk3L-1-2, MPI-ESM-P and HadCM3, and hence, we cannot examine their projected AIS SMB. The spatially integrated AIS SMB is projected to increase for the following 75 years (2025-2100) in both scenarios by all models. The spatially integrated AIS SMB from the best-scoring model CESM-LME (CESM1-CAM5 with LM forcing protocol) is projected to be $3107 \pm 92$ Gt yr$^{-1}$ (the associated uncertainties are $\pm 1\sigma$) for SSP2-4.5, and $3521 \pm 145$ Gt yr$^{-1}$ for SSP5-8.5 from 2070-2100. For the same time period, AIS SMB from models scoring worse than the models mean (BCC-CSM1-1, GISS-E2-R, MIROC-ESM, and MIROC-ES2L) is projected to be $2992 \pm 120$ Gt yr$^{-1}$ for SSP2-4.5, and $3216 \pm 148$ Gt yr$^{-1}$ for SSP5-8.5, which is slightly lower than the best-scoring model.

In terms of trends, all models project positive SMB trends in all scenarios. For CESM1-CAM5, SMB is projected to have a mean trend of $5.2 \pm 0.4$ Gt yr$^{-2}$ for SSP2-4.5, and $13 \pm 0.5$ Gt yr$^{-2}$ for SSP5-8.5, while, in the worst-scoring models, the AIS SMB mean trend is projected to be at $2.5 \pm 0.6$ Gt yr$^{-2}$ for SSP2-4.5, and $6.8 \pm 0.7$ Gt yr$^{-2}$ for SSP5-8.5. The best-scoring model suggests stronger SMB means and trends with lower uncertainties for both scenarios compared with the worst-scoring models.

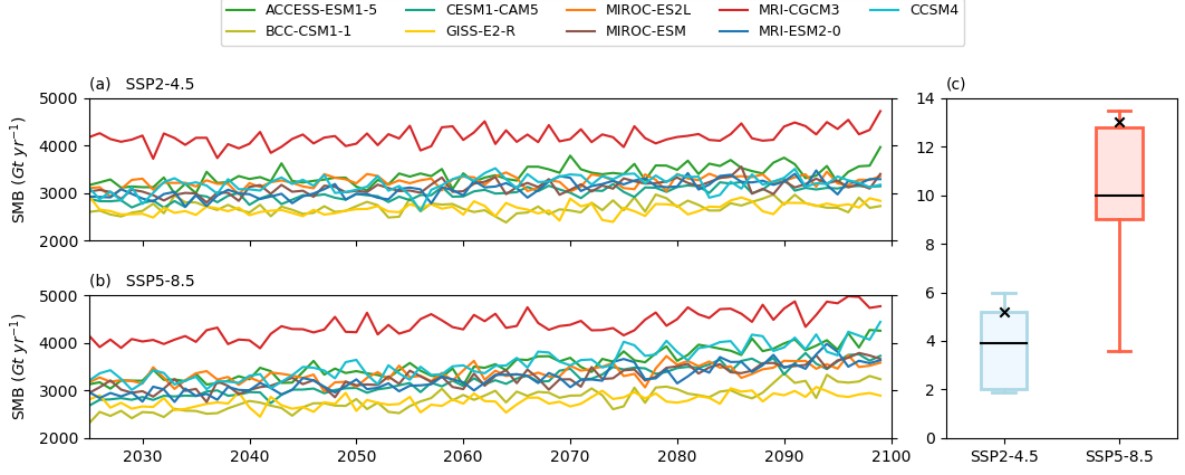

**Figure 11.** Time series of modelled spatially integrated AIS SMB projections for two scenarios: (a) SSP2-4.5 and (b) SSP5-8.5. (c) Box plots of the linear trend in spatially integrated AIS SMB from 2025-2100. The black cross denote the best overall scoring model (CESM1-CAM5).

Figure 12 shows the projected regional changes in SMB over the AIS under the high-emission SSP5-8.5 scenario. Most of the future SMB changes are concentrated along the coast, with some smaller changes in the WAIS interior. The vast majority of coastal SMB changes are positive, with only a few models showing negative changes — notably GISS-E2-R in WL, MIROC-ES2L in VL, MIROC-ESM in WAIS and MRI-CGCM3 in WS. The ensemble mean only displays positive changes, with maxima in the AP, western WAIS, DML, and WL. The best-scoring model, CESM1-CAM5, projects positive changes along nearly all coastal regions except for VL and eastern WAIS, and shows strong agreement with ACCESS-ESM1-5 and CCSM4, two models that score slightly better than the model mean.

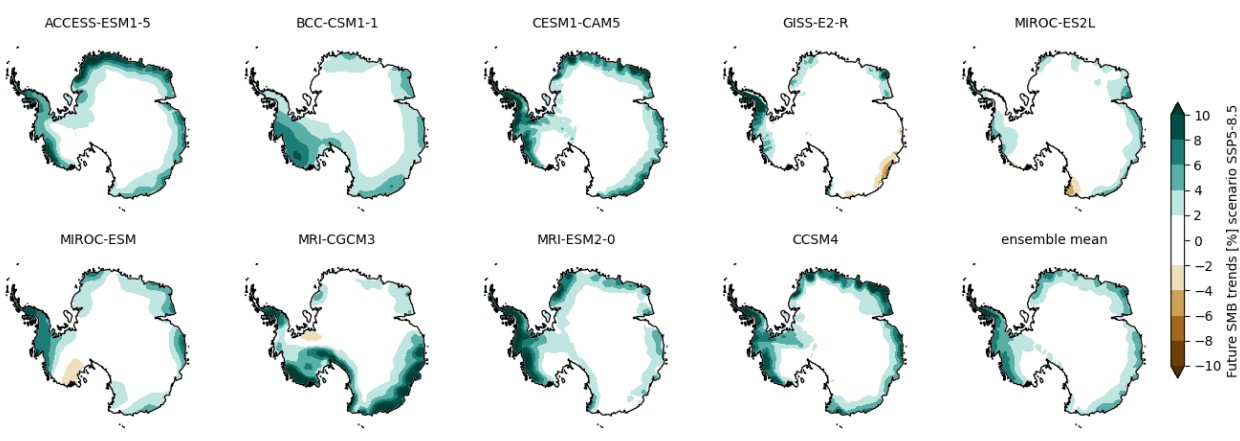

**Figure 12.** Spatial plot of the future [2025-2100] SMB change in [%] under the high-emission SSP5-8.5 scenario.

### 4.7 Relationship between climate variables

Here, we use the four best-scoring models overall (CESM-LME, CSIRO-Mk3L-1-2, MPI-ESM-P and MRI-ESM2-0) to investigate the relationship between climate variables during the LM. Figure 13 (a) shows the precipitation anomalies relative to the historical period (1850-1900 CE). While there are some regional differences, particularly in coastal East Antarctica where CESM-LME is the only one that shows no notable anomalies, most of the best-scoring models show the greatest precipitation changes along the coasts. The largest discrepancy lies in the AP region, where both MRI-ESM2-0 and MPI-ESM-P simulate positive precipitation anomalies, while both CSIRO-Mk3L-1-2 and CESM-LME simulate negative precipitation anomalies.

For the relationship between temperature and precipitation (Figure 13 (b)), local precipitation sensitivities for all four best-scoring models are not uniformly distributed. All four models agree and show a consistent local-scale positive linear relationship, and display similar regional sensitivity patterns. In general, models show lower sensitivities in the EAP and AP regions, ranging from 0 to 12 %/C° and higher sensitivities in the WS and VL regions with values that can reach up to 20 %/C°. CSIRO-Mk3L-1-2 is the only model that shows a stronger positive relationship in the interior.

Southern Ocean conditions exert a strong influence on Antarctic accumulation (Delaygue et al., 2000; Stenni et al., 2010; Lowry et al., 2019). To examine the relationship between ocean conditions and continental precipitation in the AP region, we look at the Pearson linear cross-correlation coefficients of modelled decadal SIC and precipitation (Figure 13 (c)) and of modelled decadal SST and precipitation (Figure 13 (d)). All four models agree and display similar correlation patterns in West Antarctica, with SIC and SST showing strong local spatial correlations with continental AP precipitation. The SIC-AP precipitation and the SST-AP precipitation correlations are slightly higher for MRI-ESM2-0 and CESM-LME than MPI-ESM-P and CSIRO-Mk3L-1-2. The models all exhibit high negative correlations between local SIC in the Bellingshausen and Weddell Seas and AP precipitation. MRI-ESM2-0 and CESM-LME exhibit a high positive correlation between SIC in the Amundsen Sea and precipitation in the AP region, while MPI-ESM-P and CSIRO-Mk3L-1-2 exhibit a weaker one. For SSTs, here we show that models exhibit opposite correlations with high positive correlations between local SST in the Bellingshausen and Weddell Seas and AP precipitation.

According to the four best-scoring models, regional precipitation patterns in West Antarctica are highly sensitive to local temperature and Southern Ocean conditions (SIC and SST) changes. Even though the models disagree on the precipitation changes in the AP region, there is a consistent relationship between variables, as all four models do agree in the AP region in simulating a slightly positive linear relationship with local average warming, a strong negative correlation with local SIC and a strong positive correlation with local SST.

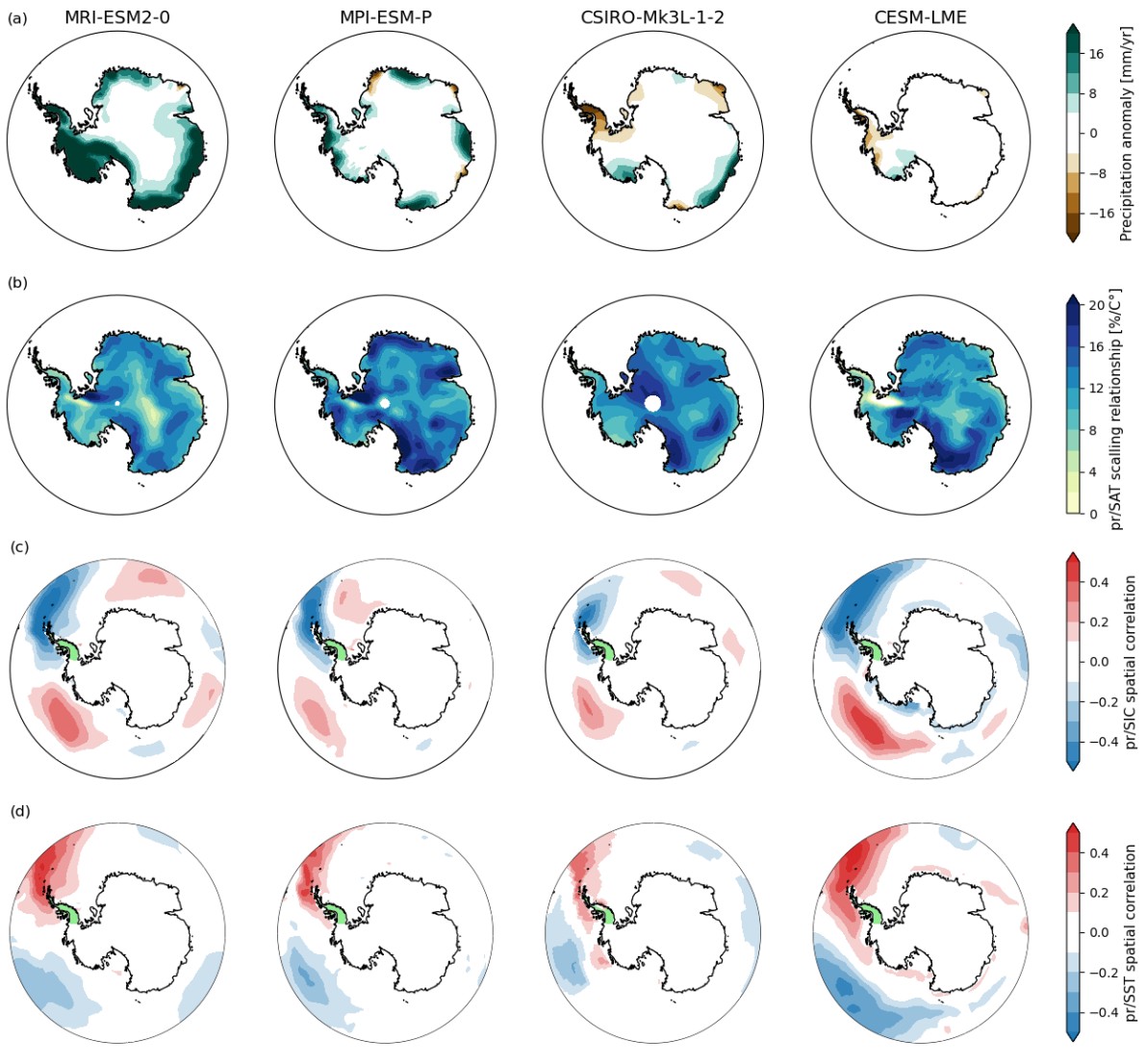

**Figure 13.** (a) Precipitation anomalies during the LM period (850-1850 CE) relative to the historical period (1850-1900 CE), (b) Spatial distribution of relative changes in precipitation rates in terms of local warming during the LM period, (c) Spatial correlation plots between precipitation in the AP region and SIC during the LM period, (d) Spatial correlation plots between precipitation in the AP region and SST during the LM period, for the best four scoring models.

## 5 Discussion

### 5.1 Regional climate features

Both Antarctic ice core records and model simulations demonstrate clear regional differences during the LM. Some models are better at representing those regional features and some models show clear regional biases. However, several elements are remarkably consistent. To start, models tend to overestimate annual snow accumulation values everywhere except in the DML and WL regions. A recent evaluation of current and projected Antarctic precipitation in CMIP5 models has shown that compared with satellite data almost all the models overestimate current Antarctic precipitation, some by more than 100% (Palerme et al., 2017). This is a recurrent issue in current models that is likely due to poor representation of coastal topography which is a significant factor in how precipitation is represented for the AIS (Genthon et al., 2009; Gorte et al., 2020).

The snow accumulation trends provide a second example of mismatch between models and reconstructions. Models simulate the incorrect magnitudes of trends and for some, they also simulate the wrong sign. Furthermore, snow accumulation can be modulated by large-scale atmospheric circulation and ocean conditions. Regional and global modes of climate variability are suggested to be the dominant controls on regional climate in Antarctica during the LM (Lüning et al., 2019). The mismatch in trends can be related to the model bias in simulating realistic patterns of decadal climate variability. Current generations of models struggle to simulate those features, especially in terms of their magnitude, spatial patterns and their sequential time development (Kravtsov et al., 2018; Mann et al., 2020).

Model-simulated SAT agrees with the reconstructed SAT and shows generalised cooling over continental Antarctica, but fails to reproduce the modest warming in the AP, and most models fail to reproduce the warming in the DML. Models that do reproduce the warming for both regions, MIROC-ESM and MIROC-ES2L, also exhibit a global warm bias. Therefore, their apparent agreement in these regions should not be interpreted as an indicator of good overall performance. Klein et al. (2019) has found the overall skill in reconstructed surface temperature based on $\delta^{18}O$ on the seven regions to be limited, but the reconstruction skill is higher and more uniform among reconstruction methods when the reconstruction targets are the bigger aggregated regions (West Antarctica, East Antarctica and Antarctica as a whole). Once averaged over these larger geographic areas, models show relatively strong agreement.

A final example of mismatch between models and reconstructions is the underestimation of Southern Ocean temperatures in models. Using SST reconstructions to constrain the model results is challenging, notably due to the large spatial gaps and temporal gaps in the records from the fact that few marine sedimentary archives have the resolution and age control necessary to reconstruct LM decadal-scale SST variability (Jones et al., 2009), and the potential for proxy-related biases (Lowry et al., 2019). We only compare four site records with three of them located relatively close to one another. Hence, there are too few Southern Ocean records that cover LM to properly evaluate models on their ability to capture SST trends. This study would be more robust with an overall greater spatial coverage of proxy records.

## 5.2 Overall model skill

Evaluations of PMIP and CESM LM simulations based on four different climate variables show that no model performs equally well for all variables. In general, models are better at simulating the SAT, and are substantially poorer at simulating snow accumulation as they have no skill in reproducing trends and temporal variabilities. LM has modest trends compared to other time periods (Thomas et al., 2017); our analysis suggests that capturing regional trends of such small magnitudes that we observe in the LM is still beyond current models' ability in view of their coarse resolution, among other limitations. Models show colder SST mean values but have skill in simulating trends and variabilities. Additionally, only a handful of models show skill in simulating ENSO (Bellenger et al., 2014). Nevertheless, some models are clearly better than others at capturing LM climate.

Atmospheric and oceanic horizontal resolution and the number of vertical layers vary widely among the models. We found that models with relatively high-resolution for the atmosphere and ocean can perform equally as well as their coarse-resolution counterparts. There does not appear to be a clear relationship between horizontal resolution and model performance. There is also a lack of a clear relationship between vertical layers and model performance. For simulated snow accumulation, models participating in PMIP (but also CESM-LME) are run at insufficient resolution to provide accurate SMB estimates in the coastal regions (Lenaerts et al., 2019). To resolve the SMB component characteristics in some of the narrowest coastal regions of Antarctica with complex topography, a grid spacing of about 50 km or finer is needed (McGregor and Dix, 2008). There are two main modelling tools to improve on the relatively coarse resolution of GCMs: statistical and dynamical downscaling. Statistical downscaling is based on the use of statistical relationships between large-scale variables and local variables and has the advantage of being computationally efficient (Hernanz et al., 2023). Dynamical downscaling is based on the use of a RCM forced at its boundaries with GCM data (Lenaerts et al., 2019). Both tools enable finer resolutions, potentially improving model performance.

There are many potential sources of model bias that are beyond the scope of this study. A non-exhaustive list includes the use of different cloud physics schemes, which can strongly influence precipitation over the ice sheet (Mottram et al., 2021), the difficulty in simulating sea ice, which can affect moisture transport and precipitation on the continent (Roach et al., 2020), and the fact that models may not adequately represent weather patterns that could impact snow accumulation in certain regions of West Antarctica, potentially leading to mismatches with Holocene snow accumulation reconstructions (Fudge et al., 2016). Such polar processes remain priorities for climate model development (Eyring et al., 2019).

In this study, we compare two generations of paleo-simulations. The mean overall skill of PMIP4 models is greater than the mean overall skill of PMIP3 models, but there are more than twice as many PMIP3 models to analyse. For Antarctic climate during the historical period (1850-2000 CE), the latest generation of CMIP6 models has been shown to present no significant improvement at simulating some aspects of the modern climate with respect to CMIP5 models (Gorte et al., 2020). Similarly, for past climates, within the few specific features we looked at, there seems to be little improvement between the different generations of models. It is possible that there are potential improvements in processes that we did not examine.

Gorte et al. (2020) have evaluated CMIP5 and CMIP6 models over the historical period to look at which models capture the influence of anthropogenic warming on SMB. Although both studies share the same objective, our scoring method differs in that we consider multiple parameters important SMB as well as a longer time scale, and our model ensemble is also more limited because fewer models have run the past1000 simulations. Hence, it is useful to compare the results for models investigated by both studies. The two time periods have different forcing, which allows for a contrast between model responses to either anthropogenic or natural variability. GISS-E2-R and MPI-ESM-P are the best-scoring models in Gorte et al. (2020), while MIROC-ES2L and CCSM4 are the worst performing models. CESM1-CAM5 performed worse than their model average. For both studies, MPI-ESM-P, ACCESS-ESM1-5 and MRI-CGCM3 perform better than the model mean. Our results highlight that model evaluation studies should consider covering longer time periods for the full context of natural variability.

## 5.3 Implications for 21st-century sea level rise

Previous studies have considered the historical time period to constrain future projections of Antarctic SMB. Palerme et al. (2017) showed that models that best capture observed historical snowfall rates tended to project larger snowfall increase; although Gorte et al. (2020) found a similar increase in SMB under the high-emission scenario, their subset of best-scoring historical models suggest smaller increases. Our sample size of available models is too small to be as conclusive as those studies, but with respect to the LM, the model that performs the best across multiple SMB-relevant variables projects greater future increases in AIS SMB. To strengthen the study, we encourage more climate modelling groups to participate in experiments such as past1000 so that projections can be constrained over time scales sufficient for end-of-century projections.

A positive SMB trend means mass gain over the surface of the ice sheet, a negative contribution to the global sea level (Ligtenberg et al., 2013). According to the best-scoring model, most of the SMB change at the year 2100 is concentrated along coastal regions, with maxima in the AP, western WAIS, DML and WL. The AIS interior remains relatively unchanged with the exception of small changes in the WAIS interior. However, at present, the ongoing dynamic ice loss in West Antarctica dominates the AIS mass balance (Shepherd et al., 2018). Medley and Thomas (2019) demonstrate that the increase of snowfall over the AIS during the 20th century did not offset the ocean-driven ice mass loss and only mitigates the AIS sea level rise contribution. Some studies have projected that ice discharge from West Antarctica will continue to dominate Antarctica's sea level contribution in the future even under low-emission scenarios (Lowry et al., 2021; DeConto et al., 2021). While some parts of the AIS will likely experience mass gain by the projected increase in SMB (Winkelmann et al., 2012; Seroussi et al., 2020), this will likely not be enough to counteract the loss of mass from the marine basins of West and East Antarctica, even though the best-scoring model in our analysis shows an increasing SMB trend of $13 \pm 0.5$ Gt yr$^{-2}$ over the next century.

Although Antarctic SMB is projected to increase overall, there is still a question of how SMB changes may impact ice shelf stability in the future. Kittel et al. (2020) discuss how atmospheric warming may lead to diverging SMB responses between grounded ice and lower elevation ice shelves. Using an RCM to better represent changes in mass and energy fluxes at the surface, they find that the projected higher temperatures are likely to decrease SMB over ice shelves, mainly due to increased run-off and meltwater that can cause ice shelves to hydrofracture. Ice shelf collapse substantially increases the AIS sea level contribution in ice sheet model projections over the next three centuries (Seroussi et al., 2024). This highlights that regional

downscaling of these coarser-resolution global models is essential to fully grasp the implications of these long-term SMB processes.

## 5.4 Process understanding gained from the best scoring models

The strong AP precipitation increase, seen in two of the four best scoring models (Figure 13), is in part attributed to the local atmospheric warming —MRI-ESM2-0 is the only model that simulates consistent warming in the AP during the LM period (Figure 5). Sea ice trends have an important influence on regional precipitation variations as sea ice-free and/or warmer SSTs promote evaporation, increasing the moisture content of the atmosphere and enhancing local precipitation (Bertler et al., 2018; Lenaerts et al., 2019; Kromer and Trusel, 2023). Hence, the potential warmer SSTs and/or SIC decline in the the Bellingshausen and Weddell seas might have led to the increase in precipitation in the AP region in MRI-ESM2-0 and MPI-ESM-P models, while colder SSTs and/or greater SIC extent in the Bellingshausen and Weddell seas might have led to the decrease in precipitation in the AP region in the CSIRO-Mk3L-1-2 and CESM-LME models.

Sea ice trends can be driven by factors other than large-scale atmospheric circulation modes, but Crosta et al. (2021) suggest that natural variability has played a crucial role over the last 2000 years, with the Southern Annular Mode (SAM) and ENSO believed to be driving regional climate heterogeneity for sea ice and sea surface temperature (SST) in the Southern Ocean. Those two modes wield their influence on West Antarctica by directly influencing the Amundsen Sea Low (ASL). Figure 13 (c) and (d) can be interpreted as the ASL affecting the sea ice and precipitation rate for the AP. The model with the best score for AP snow accumulation (MIROC-ES2L) also has the best score for ENSO. CESM-LME also ranks highly in terms of ENSO and the AP snow accumulation. However, CSIRO-Mk3L-1-2 scores poorly in terms of ENSO, but performs well with respect to AP snow accumulation.

Processes other than ENSO also impact how models simulate SAM and the ASL, which can impact precipitation change in the AP. While we do not investigate the impact of short-term fluctuations in precipitation in this study, this may contribute to the discrepancy in the AP region. Extreme precipitation events, such as atmospheric rivers (ARs), defined as long and narrow band of water vapour transport in the atmosphere (Gorodetskaya et al., 2014), have been shown to impact the AIS mass balance (Wille et al., 2025). While occurring relatively rarely over Antarctica — only a few days per year — ARs transport large amounts of moisture from the mid- to high-latitudes and can lead to extreme precipitation and surface melt events (Wille et al., 2021). Model differences in the simulation of heat and moisture transport from mid-latitudes to the Antarctic continent could potentially be behind the discrepancy in the AP region, which is particularly impacted by these processes, and needs to be further assessed.

## 6 Conclusions

The goal of this study is to provide a quantitative evaluation of GCMs in simulating LM regional climate changes in Antarctica. We assess model performance with regard to the output most relevant to AIS SMB, including snow accumulation, SAT, SST and Niño 3.4 index. The multi-parameter score used in this study is an indication of the model's performance in comparison with

other models and is designed as a guide for choosing which GCMs best represent LM AIS SMB. We apply a similar scoring method to Gorte et al. (2020) for our time series variables, as having several criteria for each variable limits the possibility that models are recreating one aspect well for the wrong reasons. Those criteria were originally suited to gauge model performance for capturing AIS SMB only, but they are also applicable to the wider range of climate variables that we consider in this study (snow accumulation, SAT and SST). For scoring the Niño 3.4 index, we evaluate whether models simulate a similar number of El Niño and La Niña events over a given time period.

CESM-LME mean is the best overall scoring model. CESM-LME is an ensemble mean composed of 13 individual members but presents very little internal variability, meaning if we were to look at only a single member, CESM-LME would still rank as the best overall scoring model. It shows strength in simulating SAT, snow accumulation and Niño 3.4 index while performing better than the average mean in simulated SST. Out of all the models studied here, CESM-LME mean is the recommended choice for forcing RCMs over Antarctica for the Last Millennium.

In general, the models show poor skill in simulating regional snow accumulation. They tend to overestimate accumulation in the WAIS, AP, WS, VL and EAP, while showing strong discrepancies with reconstructions of accumulation trends and temporal variability. The best performing model in terms of snow accumulation, CSIRO-Mk3l-1-2, shows the greatest skill in simulating accumulation mean value over West Antarctica (AP and WAIS) and the EAP, but does not capture the accumulation trends and temporal variability in every Antarctic region. MIROC-ES2L shows regional biases in the WAIS, VL and EAP regions.

Regional SATs reconstructed from the proxy record are reasonably captured by the GCMs in this study. The models are relatively consistent in displaying the modest broad-scale cooling trend over most of continental Antarctica, but fail at capturing the modest warming in the AP and DML. The exceptions are MIROC-ESM and MRI-ESM2-0, which both show an overall warm bias.

The models display a cool bias in simulating Southern Ocean SST. ACCESS-ESM1-5 and MRI-CGCM3 are able to capture consistent mean, trends and temporal variability values in all but one proxy record site. For the site on the west coast of the AP, representing SST over the austral spring, all models simulate SST at freezing temperature, suggesting that there is persistent spring sea ice cover at that location, in contrast to the proxy record, which indicates sea ice only towards the end of the LM.

The greatest model-proxy mismatch occurs in simulating the Niño 3.4 index, where only three models, MRI-ESM2-0, MIROC-ES2L and the CESM-LME mean simulate a relatively similar number of El Niño and La Niña events to the reconstruction. All of the remaining models fail to simulate realistic ENSO behaviour. These results are not surprising considering some GCMs have been demonstrated to struggle with representing ENSO (Bellenger et al., 2014).

For the models where future scenarios were available, they all show an increase in the spatially integrated AIS SMB by the end of the 21st century. The model that performs the best in simulating regional climate features over the LM and its range of natural variability implies that increases in SMB will more strongly mitigate future dynamic ice loss and sea level rise contribution from the AIS.

Our results provide some sobering evidence of the limits of the current generation of models in their ability to properly simulate regional LM climate features over Antarctica. Given the limited number of models and proxy records, it remains challenging to assess the precise reasons for the regional model/proxy mismatches, and further investigation is required. The

community would be well served by additional models participating in the past1000 experiments and better proxy spatial coverage overall, but more importantly in the WS and EAP. Although there can be large uncertainties and model biases in some cases, the PMIP past1000 models are beneficial for investigating long-term climate variability in this region. Notably, our quantitative evaluation serves as a guide for the selection of GCM forcings for model weighting of future projections,

dynamical downscaling and/or statistical downscaling.

*Data availability.* PMIP outputs are available for download from the World Climate Research Programme at https://aims2.llnl.gov/search/cmip6/. CESM-LME outputs are available for download from https://www.earthsystemgrid.org/dataset/ucar.cgd.ccsm4.cesmLME.html. The snow accumulation reconstructions data are available from the UK Polar Data Centre at https://data.bas.ac.uk/full-record.php?id=GB/NERC/BASPDC/01052. The SAT reconstructions are available in the NOAA World Data Center for Paleoclimatology (WDC Paleo) at https:

500  //www.ncei.noaa.gov/access/paleo-search/study/22589. The SST reconstructions are likewise available in the NOAA WDC Paleo at https://www.ncei.noaa.gov/access/paleo-search/study/18718. The Niño 3.4 index reconstruction is likewise available in the NOAA WDC Paleo at https://www.ncei.noaa.gov/access/paleo-search/study/8704.

*Author contributions.* VC conducted the analysis and led the writing of the manuscript. All authors contributed to the study design, the interpretation of the results, and the writing of the manuscript.

*Competing interests.* The authors declare that they have no conflict of interest.

*Acknowledgements.* Vincent Charnay received support from the Antarctic Research Centre of Victoria University of Wellington. Daniel Lowry and Elizabeth Keller were supported by the New Zealand Ministry of Business, Innovation and Employment (MBIE) through the Changing Climate and Environments programme (Strategic Science Investment Fund, contract C05X1702)

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
