# Peer review of "Evaluation of regional climate features over Antarctica in the PMIP past 1000 experiment and implications for 21st-century sea level rise"

_EGUsphere, 2024_

## Referee Comment (RC1)

**Review - Evaluation of regional climate features over Antarctica in the PMIP past1000 experiment and implications for 21st-century sea level rise**

December 17, 2024

**1 Overall quality**

The topic discussed in the paper is exciting, innovative and relatively unknown, making a firm need for research like this to be published in CP. With the changes that I discuss below, I hope this work is also published in CP.

I applaud the authors for their choice of topic, the structure and writing of the paper, and the scrutinous analysis carried out. The majority of the introduction, data, method, discussion and the conclusion sections are clear, well written and easy to follow. The sheer amount of results presented is impressive. However, the fashion they are currently presented in deems to it hard to confidently understand the results section. Perhaps a more pedagogical way of conveying the results can be found, this is discussed below. The article is abundant with analysis, discussion and evaluation, meaning it is a very interesting read. This also means, however, it is lengthy. I think the article could benefit for being shortened slightly, with the most important conclusions honed in on. I feel the authors have tried to cover all discussion points - which yes is important with a model evaluation paper, but perhaps a middle ground can be reached, where the authors are slightly more ruthless, producing a more concise work which will still deliver this important subject matter.

**2 Scientific questions**

I appreciate that the authors are very aware of the limitations of working for example, with proxy data, or coarse resolution, and introduce this from the start, in the data and method sections. This, to a degree, reinforces the scientific quality of the study, that no conclusions are being based on assumptions, but rather critical analysis. However, perhaps by section 5, I began to wonder if we can pull any robust conclusions from the models/proxies? It seems, from the perspective of the authors, the majority of the data cannot be trusted. Models struggle to simulate realistic patterns of decadal climate variability, they overestimate current Antarctic precipitation buy sometimes 100%, simulate the wrong sign of snow accumulation, fail to reproduce warming in the DML and cannot be properly evaluated on their ability to capture SST trends. So considering all these limitations, why should this data be used? It is stated in line 53 "The LM is also a useful time period in which to evaluate model skill, as there is a

relative abundance of proxy data available" , so does abundance mean it can be used for the evaluation purposes the authors are aiming for?

I understand the vitality of highlighting the flaws in the proxies and models and that this is an evaluation of model performance, but I feel the authors need to include more references and more detail as to why they had the confidence to use these datasets in the first place to perform this highly important study. If the overarching conclusion of the study is that we can't yet use models to investigate paleo behaviour in this region, such as is hinted in line 332 "capturing regional trends of such small magnitudes that we observe in LM is still beyond current models' ability" then this needs to be reinforced and discussed further. This is a very important finding in terms of encouraging research development for the Southern hemisphere modeling/proxy community. As a member of the community I would like to know what is limiting these models, or what it is that we need to focus on to improve research in this area.

**2.1   Abstract**

From the abstract, I gather the conclusion of the study is PMIP past1000 models reasonably capture SATs but not reconstructed regional snow accumulation nor Niño 3.4 index, and have some skill with SSTs but a slight cold bias. But I feel the real conclusion should be what does this mean for investigating Antarctic climate. What do these findings mean in respect to your study? What does this mean in regards to the opening line of the abstract - "Surface mass balance (SMB) of the Antarctic Ice Sheet (AIS) is an important contributor to global sea level change"? Relating the findings to this subject is what makes your study innovative and exciting.

**2.2   Section 1**

From the introduction, it is not entirely clear that this is a model/data evaluation paper. The importance of the subject and the subject area is discussed well, the need for understanding paleo conditions to navigate the future unknowns is clear, but perhaps, from the tone of the introduction, I was expecting more an analysis of Antarctic paleo climate as opposed to model performance. I wish the core purpose - the evaluation - had been stated in the abstract and early on in the introduction, instead of waiting until the end of the introduction. In line 405, the authors state "The goal of this study is to provide a fair evaluation of the strengths and weaknesses of GCMs in simulating LM regional climate changes in Antarctica." This is a great and clear sentence, that in my opinion should have been used much earlier on.

**2.3   ENSO (Section 3 and 4.4)**

I don't quite understand the purpose of exploring the ENSO Index. Where in the text does it explain how exactly it affects Antarctic climate or SMB? What is the purpose of including this analysis in regards to an Antarctic paleo climate study? The only line explaining this I found was 401, connecting ENSO with sea ice.

**2.4   Section 4.1, 4.2, 4.3**

The results section starts with listing accumulation values but these lines (161-185) are hard to follow. Simply noting the values with no benchmark, does not help the reader understand how rates change. Considering Figure 2 shows the mean, trends and temporal variability values

of reconstructed and modeled time series for each ice core, plus table S1 shows details of the ice core records, is writing the various accumulation values necessary? Can this section be restructured, so the reader can easily follow which cores show which rates, why this is and how they compare? I understand that Figure 2 aims to do this, and, whilst it displays a lot of information is perhaps slightly overloaded and therefore adds to the complexity of the section. The use of notation to show which region the data sets respond to is not very pedagogic. If we are discussing regions, it is important to have some spatial context to this, or at least clearer marker areas on the figure. Figure 3 is on the other hand very easy to understand. Could these figures be interchanged so Figure 3 is introduced first? This perhaps would help the reader understand first what is a standard accumulation rate for that area, and then a break down of how the records vary within this. In the same manner, perhaps Figure 5 and Figure 4 should be switched around (and again 6 and 7), again so the reader has more clarity understand how temperatures vary per region and then per model. The values are again listed in pros form and it adds a level of confusion to the section. I want to highlight that Figure 10, the heatmap is a very clear way to deliver a summary of the vast quantities of information presented.

It benefits the study that the clear headlines from the accumulation rates, SAT and SST sections are delivered first, before delving into the point by point values. But perhaps this could be made even clearer. This would help guide the reader to the findings of the study more efficiently.

**2.5   Section 5.1**

Lines 314-320 talk about the interesting topic of how the models recreate the trends, but I wish for more discussion of literature on this section. The line 317 "they reproduce these regional trends for the wrong reason" teases a very interesting point, but then no literature backs it up. It feels almost clickbait like. What is the wrong reason? Or does it not exist and hence why there is no literature to fulfill the argument?

**3   Technical**

- Line 23 "Projections of 21st-century SMB span a large range and contain deep uncertainties (Li et al., 2023)". What does deep uncertainty mean?

- The authors tend to rely on non scientific, quite ambiguous phrases. For example; "not too different from the present, " is used often (line 51/77). Also " and by and large" (line 366). These phrases take away from the clarity of the method and results. How is it not too different? Is it similar enough to be used as an analogue? To what extent does by and large mean? I would suggest checking the paper and replacing these colloquial phrases to more scientific and suitable language.

- In methods, Figure 1 - would be very helpful with a legend to show which colors represent which regions, rather than just noted in the caption.

- I understand the authors do not want to have too many arbitrary tables, but it would be helpful to have a least of abbreviations of the Antarctic regions that the reader can easily refer back to. Currently, I had to keep going back to check lines 109-111, which felt cumbersome.

---

## Author Comment (AC1)

**Reviewer 1**

We thank the reviewer for insightful comments that have improved the manuscript. Our response to individual comments are below in blue.

**1. Overall quality**

The topic discussed in the paper is exciting, innovative and relatively unknown, making a firm need for research like this to be published in CP. With the changes that I discuss below, I hope this work is also published in CP.

I applaud the authors for their choice of topic, the structure and writing of the paper, and the scrutinous analysis carried out. The majority of the introduction, data, method, discussion and the conclusion sections are clear, well written and easy to follow. The sheer amount of results presented is impressive. However, the fashion they are currently presented in deems to it hard to confidently understand the results section. Perhaps a more pedagogical way of conveying the results can be found, this is discussed below. The article is abundant with analysis, discussion and evaluation, meaning it is a very interesting read. This also means, however, it is lengthy. I think the article could benefit for being shortened slightly, with the most important conclusions honed in on. I feel the authors have tried to cover all discussion points - which yes is important with a model evaluation paper, but perhaps a middle ground can be reached, where the authors are slightly more ruthless, producing a more concise work which will still deliver this important subject matter.

We appreciate the Reviewer for their enthusiasm and for noting the importance of the study topic, as well as their constructive comments regarding the paper structure.

**2. Scientific questions**

I appreciate that the authors are very aware of the limitations of working for example, with proxy data, or coarse resolution, and introduce this from the start, in the data and method sections. This, to a degree, reinforces the scientific quality of the study, that no conclusions are being based on assumptions, but rather critical analysis. However, perhaps by section 5, I began to wonder if we can pull any robust conclusions from the models/proxies? It seems, from the perspective of the authors, the majority of the data cannot be trusted. Models struggle to simulate realistic patterns of decadal climate variability, they overestimate current Antarctic precipitation buy sometimes 100%, simulate the wrong sign of snow accumulation, fail to reproduce warming in the DML and cannot be properly evaluated on their ability to capture SST trends. So considering all these limitations, why should this data be used? It is stated in line 53 "The LM is also a useful time period in which to evaluate model skill, as there is a relative abundance of proxy data available" , so does abundance mean it can be used for the evaluation purposes the authors are aiming for?

I understand the vitality of highlighting the flaws in the proxies and models and that this is an evaluation of model performance, but I feel the authors need to include more references and more detail as to why they had the confidence to use these datasets in the first place to perform this highly important study. If the overarching conclusion of the study is that we can't yet use models to investigate paleo behaviour in this region, such as is hinted in line 332 "capturing regional trends of such small magnitudes that we observe in LM is still beyond current models' ability" then this needs to be reinforced and discussed further. This is a very important finding in terms of encouraging research development for the Southern hemisphere modeling/proxy community. As a member of the community I would like to know what is limiting these models, or what it is that we need to focus on to improve research in this area.

This is a great point and we appreciate this constructive feedback. We agree that while it is essential to highlight the limitations, it is equally important to justify why we have the confidence to use these model outputs and datasets, and clarify what robust conclusions can be drawn despite their respective limitations. Our aim is not to suggest that models and proxies are untrustworthy, but to show how current simulations are performing. We have revised the manuscript accordingly to clarify those points.

We first will revise the introduction to add context as to what is the main purpose of performing models/proxies comparison:

"[...] External forcing, internal variability, and model structure are the main sources of General Circulation Model (GCM) uncertainty and performing a comparison with observations will not only assess model performance and identify biases but also improve confidence in future projections. Numerous studies have evaluated climate models in their ability to simulate Antarctic climate features related to SMB over the historical period to help refine future projections (e.g., Agosta et al., 2015; Palerme et al., 2016; Gorte et al., 2020). For example, Gorte et al., (2020) found that models which best captured reconstructed historical SMB, based on mean value, trend, temporal variability, and spatial distribution, tended to project smaller SMB increase by the end of the century. In contrast, Palerme et al., (2016) showed that models which compare best with observed historical snowfall tended to project larger snowfall increase into the 21st century. These different results highlight the importance of how model performance is evaluated, and the potential limit of focusing on the historical period for understanding those future long-term changes. Hence, if we want to improve confidence and predict credible end-of-century SMB, we need a longer time period to compare against. Other studies have noted the importance of going beyond the historical period and looking at past climate using proxy-based reconstructions to assess model skill (Hargreaves et al., 2013; Schmidt et al., 2014; Bracegirdle et al., 2019). Performing a model-proxy comparison provides us with an opportunity to evaluate the performance of climate models in simulating climate features over a time period that is commensurate with projected future changes (Hargreaves et al., 2013). [...]".

Having a large number of proxy data is important in evaluating the regional climate features. We will revise the sentence "The LM is also a useful time period in which to evaluate model skill, as there is a relative abundance of proxy data available" to "The relatively abundant proxy data available in some regions makes the LM a valuable period for evaluating model skills in capturing regional climate features, as this broader spatial coverage allows for a better understanding of regional trends. The uneven regional distribution of the data will allow us to constrain some regions better than others."

Additionally, we will add this sentence in the method section to highlight that we have the confidence to use these reconstructions:

"Despite uncertainties, the reconstructions offer robust and valuable information, and the method we use for the evaluation explicitly accounts for these uncertainties, supporting their use in assessing model performance (Klein et al., 2019; Gorte et al., 2020).".

Uncertainties in external forcings and model structure are potentially what limit these models. However, a detailed analysis of the reasons behind the models' limitations and the model/proxy regional mismatch are outside the scope of this study, as we are limited by a small sample of models. We do not want the main takeaway of this study to be that those models cannot be trusted, but instead, although there can be large uncertainties and model errors in some cases, we can still use them to constrain the plausible range of variability, and demonstrate the areas in which models still need refinement. We can, therefore, make suggestions for improvement and encourage further studies on the topic.

We will revise the sentence you have highlighted to:

"capturing regional trends of such small magnitudes that we observe in the LM is still beyond current models' ability in view of their coarse resolution, among other limitations.".

We will also revise and add discussion in section 5.2, to expand on sources of uncertainty in models:

"Atmospheric and oceanic horizontal resolution and the number of vertical layers vary widely among the models. We found that models with relatively high-resolution for the atmosphere and ocean can perform equally as well as their coarse-resolution counterparts. There does not appear to be a clear relationship between horizontal resolution and model performance. There is also a lack of a clear relationship between vertical layers and model performance. For simulated snow accumulation, models participating in PMIP (but also CESM-LME) are run at insufficient resolution to provide accurate SMB estimates in the coastal regions (Lenaerts et al., 2019). To resolve the SMB component characteristics in some of the narrowest coastal regions of Antarctica with complex topography, a grid spacing of about 50 km or finer is needed (Mcgregor et al., 2008). There are two main modelling tools to improve on the relatively coarse resolution of GCMs: statistical and dynamical downscaling. Statistical downscaling is based on the use of statistical relationships between large-scale variables and local variables and has the advantage of being computationally efficient (Hernanz et al., 2023). Dynamical downscaling is based on the use of a RCM forced at its boundaries with GCM data (Lenaerts et al., 2019). Both tools enable finer resolutions, potentially improving model performance.
There are many potential sources of model bias that are beyond the scope of this study. A non-exhaustive list includes the use of different cloud physics schemes, which can strongly influence precipitation over the ice sheet (Mottram et al., 2021), the difficulty in simulating sea ice, which can affect moisture transport and precipitation on the continent (Roach et al., 2020), and the fact that models may not adequately represent weather patterns that could impact snow accumulation in certain regions of West Antarctica, potentially leading to mismatches with Holocene snow accumulation reconstructions (Fudge et al., 2016). Such polar processes remain priorities for climate model development (Erying et al., 2021)".

We will also include in the section "Process understanding gained from the best scoring models" a discussion on the need to assess the representation of extreme events in models, which can drive snowfall and surface melt regional variations:

"One other possible contributor to this increase in the region could be short-term precipitation fluctuations, though we should note that, in this study, we did not assess model representation of extreme events. Extreme

precipitation events, such as atmospheric rivers (ARs), defined as long and narrow band of water vapour transport in the atmosphere (Gorodetskaya et al., 2014), have been shown to impact the AIS mass balance (Wille et al., 2025). While occurring relatively rarely over Antarctica — only a few days per year — ARs transport large amounts of moisture from the mid- to high-latitudes and can lead to extreme precipitation and surface melt events (Wille et al., 2021). Model differences in the simulation of heat and moisture transport from mid-latitudes to the Antarctic continent can also potentially be behind the discrepancy in the AP region, which is particularly impacted by these processes, and needs to be further assessed."

And, we will revise the last paragraph of the conclusion:

"Our results provide some sobering evidence of the limits of the current generation of models in their ability to properly simulate regional LM climate features over Antarctica. Given the limited number of models and proxy records, it remains challenging to assess the precise reasons for the regional model/proxy mismatches, and further investigation is required. The community would be well served by additional models participating in the past1000 experiments and better proxy spatial coverage overall, but more importantly in the WS and EAP. Although there can be large uncertainties and model biases in some cases, the PMIP past1000 models are beneficial for investigating long-term climate variability in this region. Notably, our quantitative evaluation serves as a guide for the selection of GCM forcings for model weighting of future projections, dynamical downscaling and/or statistical downscaling.".

**2.1. Abstract**
From the abstract, I gather the conclusion of the study is PMIP past1000 models reasonably capture SATs but not reconstructed regional snow accumulation nor Niño 3.4 index, and have some skill with SSTs but a slight cold bias. But I feel the real conclusion should be what does this mean for investigating Antarctic climate. What do these findings mean in respect to your study? What does this mean in regards to the opening line of the abstract - "Surface mass balance (SMB) of the Antarctic Ice Sheet (AIS) is an important contributor to global sea level change"? Relating the findings to this subject is what makes your study innovative and exciting.

We appreciate this constructive feedback, and accordingly will revise the abstract to clarify the core purpose of this study:

"Surface mass balance (SMB) of the Antarctic Ice Sheet (AIS) is an important contributor to global sea level change. We look to the Last Millennium (850-1850 CE) as a period of relative climate stability to understand what processes control natural variability in SMB, distinct from anthropogenic warming. With evidence for large regional differences in climate and SMB trends over millennial timescales from ice core proxy records, model simulations need to be validated over long timescales to assess if they capture those regional variations. In this study, we provide a quantitative evaluation of paleo-simulations in simulating Last Millennium regional climate changes in Antarctica. We evaluate model performance by comparing available Paleoclimate Modelling Intercomparison Project (PMIP) past1000 models and the CESM Last Millennium Ensemble (CESM-LME) to four sets of Last Millennium Antarctic proxy-based reconstructions that are most relevant to the SMB: snow accumulation, surface air temperature (SAT), sea surface temperature (SST) and Niño 3.4 index, using a multi-parameter scoring method. Our results show that no single model performs consistently well across all variables. Models have reasonable strength in capturing SATs and SSTs, while

showing strong biases for both snow accumulation and the Niño 3.4 index. The best-performing model, CESM-LME, predicts higher SMB by 2100, which implies stronger mitigation of the projected dynamic ice loss contribution of the AIS to sea level rise."

We will also add an expanded discussion on future implications. Please refer to our response to Reviewer 2 for details on the proposed changes.

**2.2. Section 1**

From the introduction, it is not entirely clear that this is a model/data evaluation paper. The importance of the subject and the subject area is discussed well, the need for understanding paleo conditions to navigate the future unknowns is clear, but perhaps, from the tone of the introduction, I was expecting more an analysis of Antarctic paleo climate as opposed to model performance. I wish the core purpose - the evaluation - had been stated in the abstract and early on in the introduction, instead of waiting until the end of the introduction. In line 405, the authors state "The goal of this study is to provide a fair evaluation of the strengths and weaknesses of GCMs in simulating LM regional climate changes in Antarctica." This is a great and clear sentence, that in my opinion should have been used much earlier on.

We will revise the introduction to introduce the core focus of the study sooner, following this structure: (1) the importance of SMB, (2) model evaluation over past period, (3) context from the Last Millennium, (4) the study's aims. Please refer to our response to comment "2. Scientific questions" for details on the proposed changes.

**2.3. ENSO (Section 3 and 4.4)**

I don't quite understand the purpose of exploring the ENSO Index. Where in the text does it explain how exactly it affects Antarctic climate or SMB? What is the purpose of including this analysis in regards to an Antarctic paleo climate study? The only line explaining this I found was 401, connecting ENSO with sea ice.

We agree with the reviewer that information regarding why we are interested in evaluating model skill with respect to ENSO is lacking. In general, both the Southern Hemisphere modes of climate variability and teleconnections arising from Northern Hemisphere modes of climate variability are quite important in influencing the regional climate trends. Hence, in order to properly evaluate model skills, we require a variable representing those modes of climate variability. We plan to revise section 2.2 to clarify this point:

"Teleconnections arising from the El Niño-Southern Oscillation (ENSO) play a crucial role in shaping recent Antarctic climate trends and SMB (Lüning et al., 2019). ENSO exercises an influence on Antarctic climate by weakening or strengthening the Amundsen Sea Low, depending on its phase, which directly influences the atmospheric moisture over West Antarctica and, subsequently, the amount of precipitation (Ding et al., 2011; Clem et al., 2018). As a result, we include the ENSO index in the scoring."

**2.4. Section 4.1, 4.2, 4.3**

The results section starts with listing accumulation values but these lines (161-185) are hard to follow. Simply noting the values with no benchmark, does not help the reader understand how rates change. Considering Figure 2 shows the mean, trends and temporal variability values of reconstructed and modeled time series for each ice core, plus table S1 shows details of the ice core records, is writing the various accumulation

values necessary? Can this section be restructured, so the reader can easily follow which cores show which rates, why this is and how they compare? I understand that Figure 2 aims to do this, and, whilst it displays a lot of information is perhaps slightly overloaded and therefore adds to the complexity of the section. The use of notation to show which region the data sets respond to is not very pedagogic. If we are discussing regions, it is important to have some spatial context to this, or at least clearer marker areas on the figure. Figure 3 is on the other hand very easy to understand. Could these figures be interchanged so Figure 3 is introduced first? This perhaps would help the reader understand first what is a standard accumulation rate for that area, and then a break down of how the records vary within this. In the same manner, perhaps Figure 5 and Figure 4 should be switched around (and again 6 and 7), again so the reader has more clarity understand how temperatures vary per region and then per model. The values are again listed in pros form and it adds a level of confusion to the section. I want to highlight that Figure 10, the heatmap is a very clear way to deliver a summary of the vast quantities of information presented.

It benefits the study that the clear headlines from the accumulation rates, SAT and SST sections are delivered first, before delving into the point by point values. But perhaps this could be made even clearer. This would help guide the reader to the findings of the study more efficiently.

This is a good suggestion. Accordingly, we will revise the figures to introduce the regional map scores first and then the time series. We agree that listing the values for each variable is difficult to follow. As a result, we plan to remove these lists from the main text, as the values are still clearly shown in Figure 2 (now re-named Figure 3). We will also add, as suggested, clear headlines for each subsection.

**2.5. Section 5.1**

Lines 314-320 talk about the interesting topic of how the models recreate the trends, but I wish for more discussion of literature on this section. The line 317 "they reproduce these regional trends for the wrong reason" teases a very interesting point, but then no literature backs it up. It feels almost clickbait like. What is the wrong reason? Or does it not exist and hence why there is no literature to fulfill the argument?

We agree with the reviewer. This is complicated because it is difficult to determine whether or not the models that reproduce correct regional trends are accurately representing all of the processes. The reason why we used the term "wrong reason" is because both models show a global warm bias, and the fact that they are the only models that show the warming in the two regions (AP and DML) should not be indicative of their good performance in those two regions. Our wording did not make that point clear, as the reviewer has suggested. We will clarify this as follows:

"Models that do reproduce the warming for both regions, MIROC-ESM and MIROC-ES2L, also exhibit a global warm bias. Therefore, their apparent agreement in these regions should not be interpreted as an indicator of good overall performance."

**3. Technical**

- Line 23 "Projections of 21st-century SMB span a large range and contain deep uncertainties (Li et al., 2023)". What does deep uncertainty mean?

  We will revise this sentence for clarity:

"Projections of 21st-century SMB span a large range and involve uncertainties derived from insufficient understanding of processes important for polar climate and structural differences among climate models (Li et al., 2023)."

- The authors tend to rely on non scientific, quite ambiguous phrases. For example; "not too different from the present, " is used often (line 51/77). Also " and by and large" (line 366). These phrases take away from the clarity of the method and results. How is it not too different? Is it similar enough to be used as an analogue? To what extent does by and large mean? I would suggest checking the paper and replacing these colloquial phrases to more scientific and suitable language.

  "not too different from the present" will be changed to "conditions resembling those of the current climate". We will remove the "and by and large", the sentence is more suitable without it.

- In methods, Figure 1 - would be very helpful with a legend to show which colors represent which regions, rather than just noted in the caption.

- I understand the authors do not want to have too many arbitrary tables, but it would be helpful to have a least of abbreviations of the Antarctic regions that the reader can easily refer back to. Currently, I had to keep going back to check lines 109-111, which felt cumbersome.

  We will update Figure 1 to include a legend showing which colours represent which regions along with their abbreviations.

Figure 1

[Figure]

***Figure 1:*** *(a) Locations of ice core sites with reconstructed SAT (black crosses) and snow accumulation (blue dots) (Thomas et al., 2017; Stenni et al., 2017) and the SAT regional boundaries from (Stenni et al., 2017) used in this study. (b) Sediment core locations for the Southern Ocean sea surface temperature reconstructions for annual (blue) and seasonally averaged over the austral spring (orange) (PAGES2k, 2013).*

---

## Author Comment (AC2)

**Reviewer 2**

We thank the reviewer for insightful comments that have improved the manuscripts. Our response to individual comments are below in blue.

The authors present a comprehensive and well-executed evaluation of regional climate features—such as snow accumulation, surface air temperature (SAT), sea surface temperature (SST), and the ENSO index—that influence surface mass balance (SMB) over Antarctica. The comparison across multiple PMIP past1000 models and CESM-LME, alongside proxy reconstructions, is thorough and highlights important mismatches between models and observations. A key takeaway is that no single model performs consistently well across all variables, and even the best-performing model (CESM-LME) only marginally outperforms others in projecting future SMB increases.

Overall, the manuscript is well written, and I don't have any major concerns.

We thank the Reviewer for their positive comments.

Minor comment: I believe the manuscript would benefit from a clearer articulation of the scientific implications of its findings. While the motivation to understand SMB variability is well established in the introduction, the discussion section could more explicitly address how the model evaluation enhances our understanding of SMB and its relevance for future sea level projections. For instance, the abstract notes that CESM-LME predicts higher SMB by 2100, but the implications of this projection are not explored. To what extent does an increase in SMB contribute to sea level rise? These are critical questions that would help contextualize the study's broader significance. Based on the title of the paper, I was expecting a stronger emphasis on these implications. At present, the manuscript is heavily focused on model evaluation (which is good), but the connection to the larger scientific or societal relevance —particularly in the context of sea level rise— is underdeveloped.

We agree with the reviewer that providing a more detailed discussion of the scientific implications of increased Antarctic SMB for future sea level rise would improve the manuscript. To address this comment we will include a new figure in the results section showing the spatial pattern of SMB changes in an SSP5-8.5 high-emissions scenario. This will allow us to discuss Antarctic mass balance from a regional perspective:

[revised manuscript text omitted]

We will also add a few sentences in the conclusions with respect to these future implications:

"For the models where future scenarios were available, they all show an increase in the spatially integrated AIS SMB by the end of the 21st century. The model that performs the best in simulating regional climate features over the LM and its range of natural variability implies that increases in SMB will more strongly mitigate future dynamic ice loss and sea level rise contribution from the AIS.".

Line 361–363: Not sure if the statement "all models generally agree" is true here. For example, CESM-LME shows no change in coastal East Antarctica, while other models show notable anomalies. Consider softening this claim.

We will revise this sentence accordingly:

"While there are some regional differences, particularly in coastal East Antarctica where CESM-LME is the only one that shows no notable anomalies, most of the best-scoring models show the greatest precipitation changes along the coasts."

Line 372–373: Again, all models mostly agree for West Antarctica, but not for East Antarctica.

We will revise that sentence.

"All four models agree and display similar correlation patterns in West Antarctica, ..."

Line 378: Consider removing the parentheses around "Bellingshausen and Weddell Seas".

We will remove the parentheses.

"For SSTs, here we show that models exhibit opposite correlations with high positive correlations between local SST in the Bellingshausen and Weddell Seas and AP precipitation."

Line 381: ".......changes in the West Antarctica"

In a revised version, this will be added.

"According to the four best-scoring models, regional precipitation patterns in West Antarctica are highly sensitive to local temperature and Southern Ocean conditions (SIC and SST) changes."

Line 381–384: This point appears to repeat content from lines 362–363.

In a revised version, we will remove this repeated content.

Section 5.3: I think much of this section could be moved to results.

We agree with the reviewer that much of this section is better placed in the Results. We plan to name this section "4.7 Relationship between climate variables". Part of the section that will not be moved and is related to the relative importance of decadal climate variability in driving the precipitation will be kept under the same discussion name, "Process understanding gained from the best scoring models".